# Discovery of a Ni²⁺-dependent heterohexameric metformin hydrolase

Tao Li [1], Zhi-Jing Xu[1], Shu-Ting Zhang [1], Jia Xu[1], Piaopiao Pan [1] & Ning-Yi Zhou [1] ✉

The biguanide drug metformin is a first-line blood glucose-lowering medication for type 2 diabetes, leading to its presence in the global environment. However, little is known about the fate of metformin by microbial catabolism. Here, we characterize a Ni²⁺-dependent heterohexameric enzyme (MetCaCb) from the ureohydrolase superfamily, catalyzing the hydrolysis of metformin into guanylurea and dimethylamine. Either subunit alone is catalytically inactive, but together they work as an active enzyme highly specific for metformin. The crystal structure of the MetCaCb complex shows the coordination of the binuclear metal cluster only in MetCa, with MetCb as a protein binder of its active cognate. An in-silico search and functional assay discover a group of MetCaCb-like protein pairs exhibiting metformin hydrolase activity in the environment. Our findings not only establish the genetic and biochemical foundation for metformin catabolism but also provide additional insights into the adaption of the ancient enzymes toward newly occurred substrate.

Active pharmaceutical ingredients (APIs) are specifically designed to benefit humans or livestock. These compounds are continuously discharged into the environment, inadvertently becoming a source of contamination in diverse ecosystems[1,2], and therefore eliciting underestimated effects by interacting with animals, plants and microbes. Microbial catabolism of APIs has sparked significant interest, as the environmental microbial degradation of APIs offers a shield to mitigate the environmental impact of API contaminants. Additionally, gut microbes catabolizing APIs have been found to have a profound impact on drug efficacy, bioavailability, and potential side effects[3,4]. However, the microbial catabolism mechanisms for many APIs are only beginning to be uncovered.

Metformin (1,1-dimethylbiguanide) is an anthropogenic biguanide derivative with a history spanning over 100 years[5]. It is an extremely hydrophilic compound with a p$K$a value of 11.5, and usually exists as an organic cation. Metformin is known for its therapeutic effects on type 2 diabetes (T2D) and has potential effects against cancer, cardiovascular diseases, aging, and inflammation[6]. A substantial amount of metformin is consumed each year due to its use at a dose of 0.5–2.5 g daily by T2D individuals, estimated to grow to 783 million people by 2045[7]. However, the orally administered metformin is mainly excreted by the human body as an unmetabolized prototype compound, making it one of the most prevalent APIs in the environment globally[2,8,9].

Evidence is now available on the metformin-microbiota interactions[6,10,11] and biodegradability of metformin[12]. However, whether gut microbiota can catabolize metformin and thereby affect metformin-microbiota interactions remains an unanswered question. Nonetheless, metformin biodegradation has been identified in many soil and aquatic bacterial consortia over the past decade[13,14]. The analysis of its catabolic intermediates has unveiled several potential catabolic pathways, with guanylurea emerging as the most commonly detected metabolite accumulating in the environment[12]. Only recently the genes responsible for guanylurea catabolism were identified in *Pseudomonas* spp.[15,16], in which a hydrolase encoded by *guuH* initiated guanylurea catabolism with the production of guanidine and ammonia[15]. However, to our knowledge, enzymes mediating the cleavage of metformin have not been characterized.

Recently, we identified a metformin utilizer, *Aminobacter* sp. strain NyZ550, that catabolized metformin through hydrolysis to produce guanylurea and dimethylamine[17]. Meanwhile several other metformin utilizers were also isolated[16,18], providing further support for our predicted metformin catabolic pathway. However, the

[1]State Key Laboratory of Microbial Metabolism, Joint International Research Laboratory of Metabolic and Developmental Sciences, and School of Life Sciences and Biotechnology, Shanghai Jiao Tong University, 200240 Shanghai, China. ✉e-mail: ningyi.zhou@sjtu.edu.cn

molecular and biochemical mechanism of metformin catabolism in these strains remains elusive. In this study, we report that two paired arginase genes encoding a two-subunit hydrolase are responsible for metformin catabolism in *Aminobacter* sp. strain NyZ550. Together with crystal structure and mutagenesis studies, we demonstrated that MetCa and MetCb composed a distinct heterohexameric enzyme through intermolecular interactions. We also revealed the divergent functional evolution of the proteins from the ancient arginase family, in which MetCa retained a metal-containing catalytic subunit but MetCb evolved to be a pseudoenzyme as metal-free binding partner.

## Results

### A $Ni^{2+}$-dependent metformin hydrolase

When examining the genomes of several metformin utilizers recently characterized[16–19], a gene cluster with seven genes (designated *metR2TR1ABCaCb*) was found to be present in all of them (Supplementary Fig. 1). This gene cluster consists of two transcriptional regulators belonging to the AcrR family regulator (MetR1) and MerR family regulator (MetR2), a nucleobase-cation-symport-1 (NCS1) like importer (MetT), two genes annotated as Ni/Fe hydrogenase nickel incorporation-associated proteins (MetA and MetB), as well as two arginase/agmatinase family proteins (MetCa and MetCb). The differential omics analyzes revealed that the expression levels of the genes *metCaCb* encoding arginase family proteins are upregulated under metformin treatment condition[17,18]. Arginase family enzymes are metalloenzymes, usually preferring $Mn^{2+}$ for the catalysis of cleavage of C−N bonds[20]. In this regard, the *met* gene cluster encoding metallohydrolases, metallochaperones, a putative uptake system, and transcriptional regulators might encode metformin catabolism (Fig. 1a).

When the two arginase family proteins MetCaCb from strain NyZ550 were coexpressed in *E. coli*, low activity was detected in soluble protein extracts without the addition of transition metal salts (Fig. 1b). This activity was absent in extracts from *E. coli* cells carrying an empty vector. The addition of $Ni^{2+}$ to the cell extracts strongly increased the activity and $Co^{2+}$ slightly increased the activity, but the activity was inhibited by the addition of $Zn^{2+}$ and $Cu^{2+}$. Coexpression of the two metal chaperones MetAB with MetCaCb had no significant influence on the activity (Fig. 1b). Additionally, when MetCa or MetCb was individually expressed in the absence or presence of MetAB, no metformin hydrolase activity was detected in the soluble cell extracts (Supplementary Fig. 2a). It was observed that MetCa had low solubility when expressed alone in *E. coli*, while coexpression with MetCb improved its solubility. To rule out the possibility that the low solubility of MetCa in the cell extracts led to a failure to detect activity, the recombinant MetCa and MetCb were purified individually for activity assays. As shown in Fig. 1c, neither protein exhibited activity in vitro, even in the presence of excess $Ni^{2+}$. Likewise, no activity was observed when equal amounts of MetCa and MetCb were mixed and pre-incubated at 30 °C for 1 h before the addition of 10 mM metformin. However, activity appeared when MetCa and MetCb were coexpressed and purified (Fig. 1c, d; and see below), suggesting that *metCa* and *metCb* together encode metformin hydrolase activity.

Since six additional genes encoding arginase family proteins were present in the genome of strain NyZ550, gene deletion was performed to verify whether the metformin hydrolase activity of MetCaCb was exclusive to strain NyZ550. With the deletion of coding sequence of either *metCa* or *metCb* following a two-step homologous recombination strategy, the mutants were no longer able to utilize metformin as the sole carbon and nitrogen source for growth, whereas their capacity to

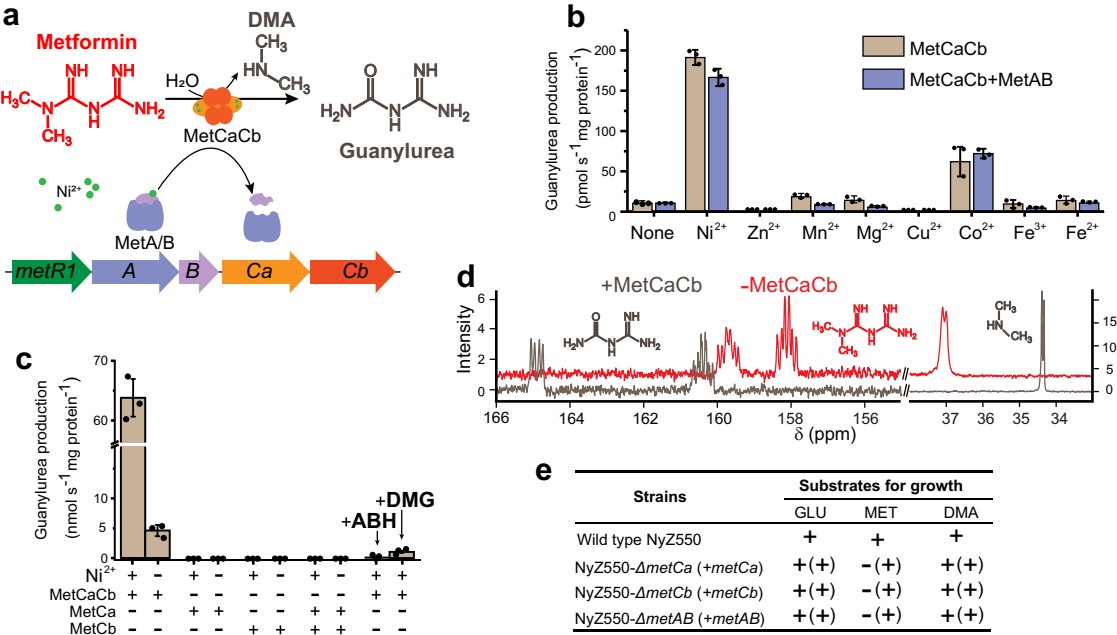

**Fig. 1 | MetCa and MetCb together function as a $Ni^{2+}$-dependent metformin hydrolase. a** Schematic representation of metformin hydrolysis and the putative genes involved. DMA, dimethylamine. **b** Conversion of metformin by cell extracts of *E. coli* expressing MetCaCb alone or together with MetAB in the presence of various transition metal ions. The cell extracts were incubated with 10 mM metformin and 0.2 mM of the indicated metal ions, and guanylurea production was measured after 12 h of incubation. **c** Metformin hydrolase assays with purified MetCa, MetCb and co-purified MetCaCb. The components of the reaction mixtures are indicated below each column. The arginase inhibitor 2(S)-amino-6-boronohexonic acid (ABH) and the nickel (Ni)-specific chelator dimethylglyoxime (DMG) were tested for their effects on the activity of MetCaCb. All assays in (**b**) and

(**c**) were performed in three independent preparations. Data are presented as mean values ± SD. **d** $^{13}C$-NMR spectra of $^{13}C^{15}N$-labeled metformin (10 mM) reaction mixtures with or without recombinant MetCaCb. **e** Growth phenotypes of strain NyZ550 and its gene-knockout variants on 5 mM glucose (GLU), metformin (MET), and dimethylamine (DMA). The respective gene complementation and their growth phenotypes are indicated within parentheses. MET and DMA served as the sole sources of carbon and nitrogen, while GLU was used as the sole carbon source with ammonium sulfate as nitrogen source. The "+" means that the value of $OD_{600nm}$ was >0.1 at the stationary phase, which represents a more than 10-fold increase of the $OD_{600nm}$. The "−" means there was no evident increase of the $OD_{600nm}$ observed at an initial inoculum of 0.01.

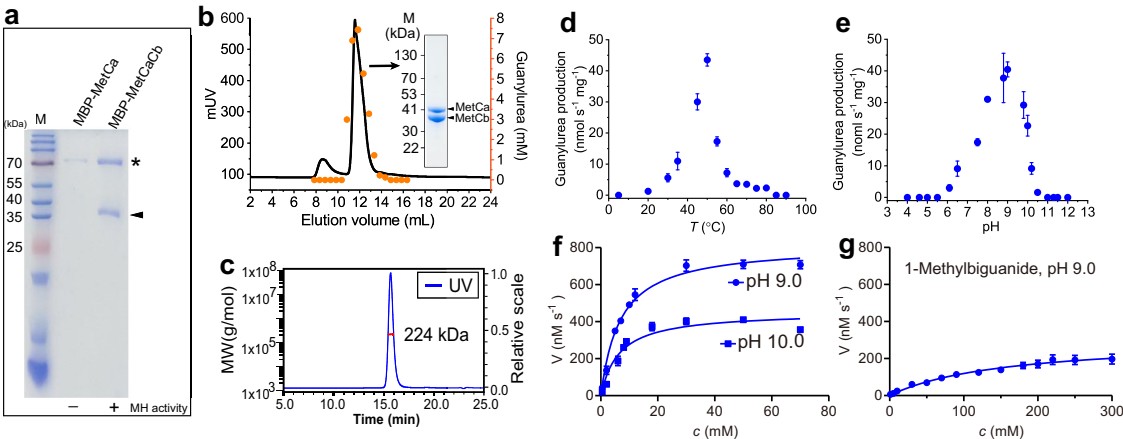

**Fig. 2 | Characterization of metformin hydrolase. a** Co-purification analysis of MetCaCb. MBP-tagged MetCa with or without tag-free MetCb were expressed in *E. coli* cells and purified using a dextrin column. The asterisk indicates the position of MBP-tagged MetCa, and the triangle indicates the position of tag-free MetCb. The metformin hydrolase (MH) activity is indicated at the bottom. Uncropped image of the gel can be found in source data file. The result shown is from a representative experiment of more than three times. **b** The size exclusion chromatography (SEC) analysis of the His-tagged MetCaCb complex. The activities of each fraction were indicated by guanylurea production within 60 min in 1-mL reaction mixture containing 5 µl of enzyme, 20 mM metformin and 0.1 mM $Ni^{2+}$. An active fraction was analyzed by SDS-PAGE (insert). The result shown is from a representative experiment of more than ten times. Uncropped image of the gel can be found in source data file. **c** Determination of the molecular weight of the active MetCaCb complex by SEC coupled with MALS. The red line represents the experimentally determined molecular weight. **d** Guanylurea production at various temperatures and pH 9.0. **e** Guanylurea production from metformin at various pH values. **f** Guanylurea production at various metformin concentrations was determined at pH 9.0 and pH 10.0. **g** Guanylurea production at various 1-methylbiguanide concentrations was determined at pH 9.0. All assays were performed in triplicate of the same preparation, and independent enzyme preparations exhibited consistent results. Data are presented as mean values ± SD. Source data are provided as a Source Data file.

grow with dimethylamine was unaffected (Fig. 1e). Moreover, a *metAB* mutant also lost the ability to grow on metformin, indicating that MetAB was required for metformin utilization in vivo (Fig. 1e). It was then concluded that MetCa and MetCb together function as a $Ni^{2+}$-dependent metformin hydrolase mediating the utilization of metformin.

## MetCa and MetCb form a heterocomplex

To examine whether MetCa and MetCb bind each other to form a heteromultimeric metformin hydrolase, MetCaCb was purified from cell extracts containing maltose-binding protein (MBP)-tagged MetCa and tag-free MetCb. MetCb was co-purified with MBP-MetCa, as evidenced by two bands visible on the SDS-PAGE, and metformin hydrolase activity was only detected in co-purified MetCaCb (Fig. 2a). This result was also repeatable when the MBP tag was replaced with a His tag purified by Ni-nitrilotriacetic acid (NTA) agarose (Fig. 2b). MetCa and MetCb comprise 357 (theoretically 40 kDa) and 348 (theoretically 38 kDa) amino acids, respectively. Size exclusion chromatography (SEC) coupled with multi-angle light scattering (MALS) analysis showed that the molecular weight of the co-purified active MetCaCb complex was determined to be 223.7 ± 0.6 kDa in solution (Fig. 2c), approximately a hexamer. These results confirmed that the interaction of MetCa with MetCb to form a heteromultimeric complex was essential for its metformin hydrolase activity.

## Characterization of the metformin hydrolase

[13]C-NMR and HPLC-MS analyzes of the reaction mixture containing the [13]C[15]N-labeled metformin as the substrate indicated that MetCaCb catalyzed the conversion of metformin into guanylurea (Fig. 1d and Supplementary Fig. 2b, c). The oxygen atom of guanylurea was derived from water, as evidenced by $H_2^{18}O$ label assays (Supplementary Fig. 2d). The other product from the hydrolysis of metformin by MetCaCb was determined to be dimethylamine, which is stoichiometrically formed with guanylurea (Supplementary Fig. 2e). Dimethylamine is a productive product used as a carbon and nitrogen resource by strain NyZ550 (Fig. 1e).

The MetCaCb's dependence on $Ni^{2+}$ was confirmed by the significant inhibition of its metformin hydrolase activity in the presence of the Ni-specific chelator dimethylglyoxime (DMG) (Fig. 1c). With the addition of $Ni^{2+}$, the metformin hydrolase MetCaCb performed under optimal reaction conditions similar to other $Mn^{2+}$-dependent arginase family enzymes[20]. The optimum temperature for the reaction was ~50 °C (Fig. 2d). MetCaCb hydrolyzed metformin most efficiently at pH 9.0 (Fig. 2e), and exhibiting a Michaelis constant ($K_m$) of 6.84 mM for metformin with a catalytic efficiency of 1.88 $mM^{-1}$ $s^{-1}$ (Fig. 2f and Supplementary Table 1). The hydrolase activity was inhibited by the arginase inhibitor 2(S)-amino-6-boronohexonic acid (ABH)[21] (Fig. 1c).

MetCaCb also catalyzed the hydrolysis of biguanide and its derivatives, including 1-butylbiguanide, 1-methylbiguanide, and phenformin with the production of guanylurea, albeit with markedly reduced activities (Supplementary Fig. 2f). The low affinity and efficiency of MetCaCb toward 1-methylbiguanide compared to that for metformin (Fig. 2g and Supplementary Table 1) was consistent with the previously observed weak growth of strain NyZ550 on 1-methylbiguanide[17]. No hydrolase activity of MetCaCb was observed toward other guanidinium moiety-containing substrates, including guanidine, dimethylguanidine, agmatine, L-arginine, guanidinobutyrate, and guanidinopropionate (Supplementary Fig. 2f).

## Architecture of the MetCaCb complex

To gain insight into the structure of metformin hydrolase and its specific binding of metformin, a crystal structure analysis of the enzyme was initiated using X-ray crystallography. The co-purified MetCaCb was subjected to SEC before crystallization, and the active fractions of the MetCaCb protein complex (Fig. 2b) were used for crystallizing by the sitting-drop method. Finally, the atomic structure of the substrate-free MetCaCb complex was resolved at a resolution of 1.84 Å (PDB NO. 8X3G). The asymmetric unit (AU) of the $P2_12_12_1$ crystal showed an uneven stoichiometry of the two subunits, which contained two MetCa (α subunit) and four MetCb (β subunit) molecules (Fig. 3a and Supplementary Fig. 3a). This uneven composition of the MetCaCb complex is also reflected by the different intensities of SDS-PAGE bands (Fig. 2b). The hexamer was composed as a centrosymmetric dimer of trimers in which a MetCa

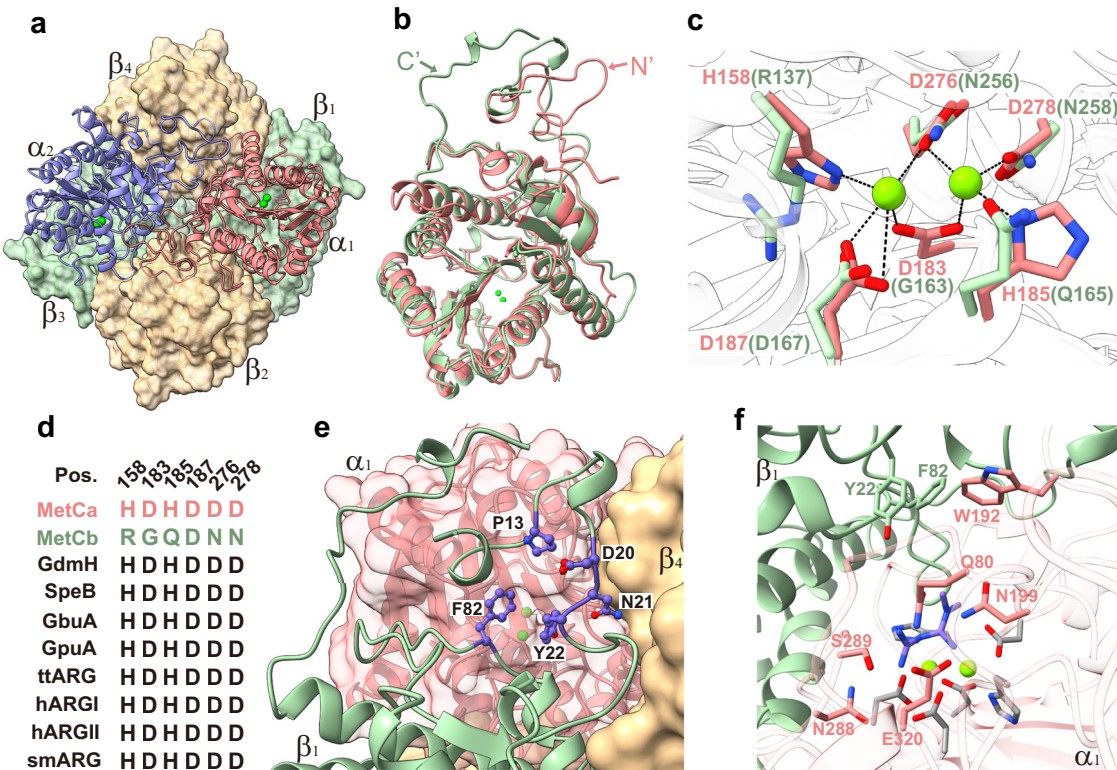

**Fig. 3 | Structural features of MetCaCb. a** The heterohexameric MetCaCb complex structure shown in ribbon (MetCa) and surface (MetCb) diagrams. Metal ions in the active sites are shown as green spheres. **b** Structural comparison of MetCa and MetCb. The atomic structures of α1 (claret) and β1(cyan) subunits are overlayed. The differential loops of MetCa or MetCb are indicated by black arrows. **c** Superposition of the active site-regions of MetCa and MetCb. Dashed black lines indicate metal coordination. **d** The conserved metal coordinating residues in representative sequences of arginase family proteins. Detailed information on the selected sequences can be found in the Methods section. Pos.: residue positions in MetCa. **e** The active site pocket in MetCa is capped by the N-terminal loop structure of MetCb. The residues around the entrance of pocket are shown as sticks. **f** Schematic drawing of the active site with a metformin molecule (purple stick) docked into it. The metal ion coordinating residues are colored in gray.

and two MetCb molecules were arranged in a head-to-tail manner (Supplementary Fig. 3). To observe the polymeric state of the Met-CaCb complex approximating physiological states, we performed single-particle cryoEM using the same sample for crystallization (Supplementary Fig. 4). Our analysis revealed a remarkable level of uniformity among the enzyme particles (Supplementary Fig. 5a, b). The resulting map presents as an asymmetric cylinder, with dimensions approximately 45 Å in radius and 80 Å in height. One side of the map exhibits tight cohesion, while the opposing side diverges to create an aperture. Subsequent fitting of the crystal structure model of the MetCaCb hexamer onto the cryoEM map provides unambiguous insights of the distribution of subunits within the enzyme: the tightly connected side corresponds to the interface where MetCa interacts, whereas the divergent region accommodates MetCb (Supplementary Fig. 5c). These findings unequivocally elucidate the hexameric configuration of the Met-CaCb complex.

MetCa and MetCb share 34% identity in their amino acid sequences, whereas their overall architectures were found to be basically identical (rms deviation of 1.94 Å for 321 Cα atoms), and exhibited the canonical three-layer alpha-beta-alpha fold of the arginase protein superfamily (Fig. 3b). The major structural differences between MetCa and MetCb lay in their N- and C-termini. MetCa had an additional N-terminal loop (T8–G26, average B-factor of 25.7) contributing to the interaction with another MetCa subunit through a hydrophobic interface (Supplementary Fig. 6a). In contrast, an extended C-terminal loop of MetCb was located on the MetCa-MetCb interface, and it also stabilized the N-terminal structure of MetCb that capped the substrate-binding cavity of MetCa (Supplementary Fig. 6b).

Another difference between MetCa and MetCb was found in the arginase family active center region. Although the corresponding regions in both MetCa and MetCb were folded in a similar conformation, only the active site of MetCa contained a bimetallic center coordinated by four Asp (D183, D187, D276 and D278) and two His (H158 and H185) residues (Fig. 3c). This metal ion coordination pattern is conserved among bacterial arginase family enzymes (Fig. 3d). A glycerol molecule from the crystal buffer was bound in the active site of MetCa and coordinated to the two metal ions (Supplementary Fig. 6c). Variants that disrupted the binuclear metal ion cluster in MetCa were introduced by substituting the conserved coordinating residues with Ala, which led to a loss of activity of the enzyme (Supplementary Table 1). In contrast, no electron density for metal ion was observed in the corresponding region of the subunit MetCb (Supplementary Fig. 6d). This was likely due to the mutations in five of the six conserved metal ion coordinating residues in MetCb (Fig. 3c, d). Analysis of the metal content by inductively coupled plasma-mass spectrometry (ICP-MS) indicated that there were 3.9 nickel atoms per $\alpha_2\beta_4$ hexamer when the enzyme was purified from *E. coli* cells grown in nickel-rich medium. In contrast, metal ions were not detected in the recombinant MetCb protein under same conditions. The results are consistent with the aforementioned observations from the protein complex structure of MetCaCb. Taken together, these results inferred MetCa as the catalytic center of this metalloenzyme.

## Active site architecture of MetCaCb

An unexpected discovery from the crystal structure of the holoenzyme is that the active site cavity within the head domain of MetCa is capped by the tail domain of a MetCb subunit (Supplementary Fig. 7a). This

configuration is distinct from typical arginase family proteins, which feature an open, solvent-exposed substrate-binding pocket[22]. The N-terminal 23 amino acids of MetCb adopted a hollowed-out structure, predicted to serve as the substrate entrance connecting to the active site cavity of MetCa (Fig. 3e). This flexible loop (average B-factor of 31.6) located at the interface of three subunits α1β1β4 or α2β2β3. Notably, the truncation of these 23 amino acids in the N-terminal of MetCb resulted in a failure to achieve soluble expression of the Met-CaCb complex, suggesting its contribution to form the hetero-multimeric protein complex.

The noncoordinating residues in the active site of MetCa were mostly hydrophilic, including Q80, N199, N288, S289, and E320 (Fig.3e and Supplementary Fig. 7b). Next, a metformin molecule was successfully docked into the metal ion-containing active center of MetCa, which revealed that metformin was accommodated in the highly negative pocket buried by the aromatic amino acids Y22 and F82 from MetCb and W192 from MetCa (Fig. 3f). The N199 and S289 residues of MetCa appeared to be unique among arginase family enzymes (Supplementary Fig. 7c). The equivalent residue of N199 in the arginase family enzymes is a conserved noncoordinating histidine which is catalytically important during the reaction process[23]. The activity of the N199H variant was significantly decreased, and the activity of the N199A variant is completely lost (Supplementary Table 1). The residue N199 likely plays a role similar to its equivalent histidine in arginases which is used for stabilizing the reaction intermediate[23]. Additionally, the substrate pocket residue S289 and E320 was also essential for metformin hydrolase activity (Supplementary Table 1).

### MetCaCb and paired arginase family proteins

Arginase family proteins are commonly known as single gene-encoding enzymes that function as monomer or homomultimer[20]. However, the heterohexameric metformin hydrolase encoded by the arginase gene pair represents a unique member in this family (Supplementary Fig. 8). To investigate the possible distribution of the MetCaCb-like arginase pairs, their homologs with at least 25% amino acid sequence identity were searched in the NCBI nr database[24]. As a result, 56 gene pairs were identified from four phyla of bacteria including *Proteobacteria* (44), *Actinomycetota* (10), *Acidobacteriota* (1) and *Planctomycetota* (1) (Supplementary Data 1). The occurrence of these paired arginases in soil, aquatic, and plant-associated bacteria underlies their environmental ubiquity. A phylogenetic tree analysis showed that MetCa and MetCb were located in two well-separated clades that evolved from a common origin (Fig. 4a). Homologs within the MetCa clade retained all six conserved metal ion coordinating residues, whereas the MetCb clade contained a group of unique arginase family proteins exhibiting three to five mutations of the six conserved coordinating residues (Fig. 4a). The genetic context analysis indicated that most encoding genes within the MetCaCb clades were found in close physical proximity to hydrogenase maturation nickel metallochaperones genes that shared homology with MetAB in this study, indicating their likely dependence on Ni²⁺ ion as well. Furthermore, there was a set of arginase pairs that contained the intact metal ion coordinating residues in both members of the pairs. Nevertheless, their specific functions, metal ion-dependence, and whether they collaborate in their enzymatic activities remain open questions.

To test whether these MetCaCb homologous pairs were functional, three codon-optimized DNA sequence pairs (AcMetCaCb, HmMetCaCb, and RsMetCaCb, as shown in Fig. 4a) from the MetCaCb clades were synthesized for the expression of recombinant proteins in *E. coli*. The α subunits within these three pairs exhibited AA identities of 93.8%, 71.2%, and 61.7% with MetCa. All the protein pairs were successfully expressed and purified to be heterohexamers as the positive control MetCaCb (Fig. 4b). In the presence of Ni²⁺, conversion of metformin into guanylurea by AcMetCaCb and HmMetCaCb was

confirmed by LC-MS. However, the hydrolase activities were approximately 20,000-fold lower in comparison to that of MetCaCb. In contrast, no detectable enzymatic activity against metformin was observed for RsMetCaCb (Fig. 4c).

## Discussion

The frequent occurrence of metformin and the accumulation of its metabolite guanylurea in the environment has raised interest in their transformation by microorganisms. The recent discovery that metformin is utilized by several bacteria isolated from different continents further supports the notion that microorganisms have developed catabolic enzymes specifically tailored to the synthetic metformin molecule. Here, we characterized a distinct bacterial enzyme capable of hydrolyzing metformin into guanylurea and dimethylamine from *Aminobacter* sp. strain NyZ550. Biochemical and structural studies demonstrated that the metformin hydrolase was a nickel metalloenzyme derived from the ancient arginase family. During revision of this manuscript, MetCaCb homologs from *Pseudomonas mendocina* sp. MET-2 were also demonstrated to be metformin hydrolase[25], which independently confirms the findings in this study.

Enzymes from the arginase family are metallohydrolases that typically use two Mn²⁺ ions to activate a water molecule for nucleophilic attack of the guanidinium group, although there are several exceptions, such as the Fe²⁺-dependent agmatinase (SpeB) from *Methanocaldococcus jannaschii*[26], Zn²⁺-dependent guanidinobutyrases from *Aspergillus niger*[27] and *Arthrobacter* sp. KUJ 8602[28], and Co²⁺-preferring arginase from *Helicobacter pylori*[29]. Notably, the guanidine hydrolase (GdmH) from *Synechocystis* sp. PCC 6803 is a Ni²⁺-dependent enzyme[30]. Both metformin and guanidine have a more resonance stabilized structure compared to the arginine, the substrate of arginase. Therefore, the nickel-dependency of metformin and guanidine hydrolases would be due to that the Lewis acidity of nickel is higher than that of manganese, which is likely required for catalyzing the hydrolysis of metformin and guanidine. Like other nickel-dependent enzymes, such as [NiFe]-hydrogenase and urease, activation of GdmH requires two nickel incorporation chaperones (GhaAB). In this study, the genes (*metAB*) encoding metal chaperones were found adjacent to *metCaCb*. It seems that they were not essential for the in vitro activity of MetCaCb (Fig. 1b). However, the inability of the *metAB* knockout strain to grow on metformin suggested that MetAB was necessary for the function of metformin hydrolase in an intracellular environment in which the concentration of nickel ion is far lower than that used in the in vitro assays.

The crystal structure and metal content analyzes were consistent with the conclusion that the MetCa subunit contained a binuclear metal center. We were able to clearly assign two metal ions to the active site of MetCa from the structure of the MetCaCb complex. The di-nickel center can be observed by providing with excess nickel ions in the growth medium. Nonetheless, the mechanism by which nickel is inserted into MetCa under physiological conditions and nickel-dependent activation of the metformin hydrolase needs further investigation. In contrast, the MetCb subunit of metformin hydrolase lost its ability to bind metal ions due to variations in the conserved coordinating residues (Fig. 3d). Notably, noncanonical arginase proteins without a binuclear metal center have been observed in both eukaryotes[31,32] and prokaryotes[33], however, their biochemical and physiological functions have been elusive. These proteins are known as pseudoenzymes that are incapable of catalyzing metal ion-dependent hydrolysis. However, their metal binding ability and hydrolase activity can be restored by reconstructing the binuclear metal center, indicating that they are evolutionarily close to the active arginase family enzymes. In this study, we uncovered a precedent in which the inactive arginase family protein MetCb mediated the hydrolase activity by interacting with its catalytically active partner MetCa, which was physiologically essential for the utilization of metformin by the strain.

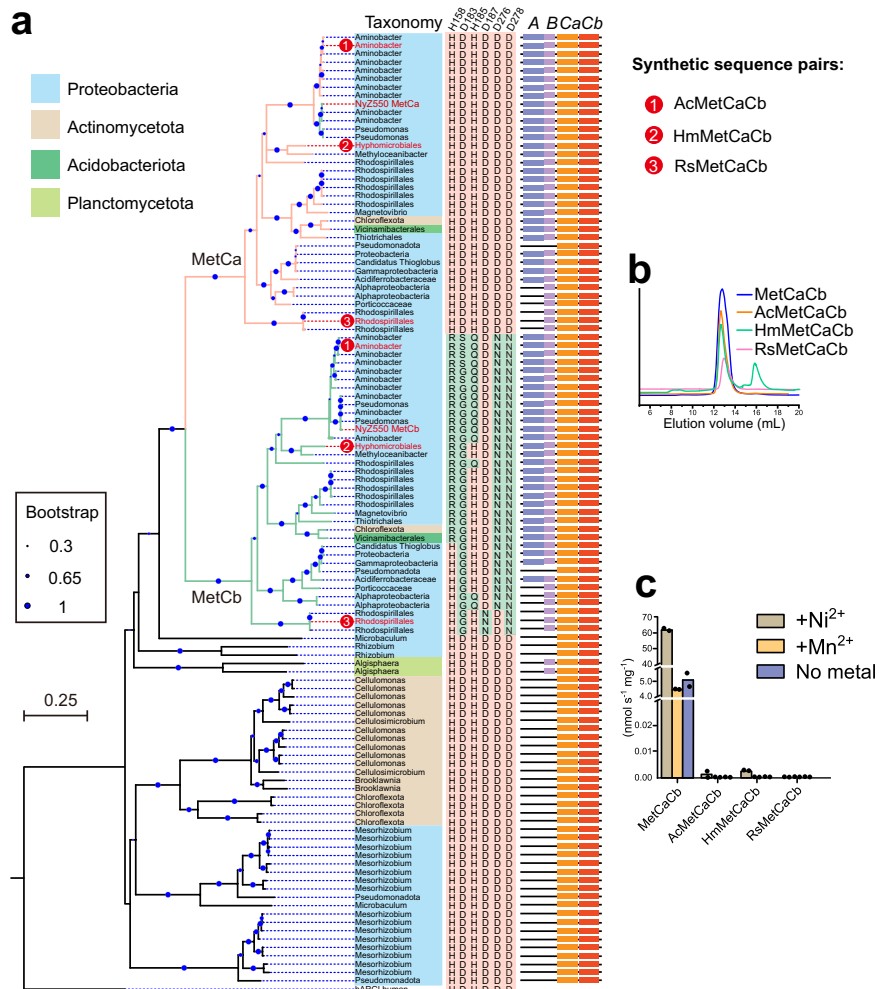

**Fig. 4 | Bioinformatic and functional analyzes of MetCaCb-like paired arginase family proteins. a** Phylogenetic tree of MetCaCb-like arginase family protein pairs. The six conserved metal ion coordinating residues for each sequence are shown on the right side, and their positions in MetCa are indicated on the top of the columns. The residues with a cyan background are mutated residues, differing from the conserved residues marked with a pink background. Human arginase I (hARGI) was used as an outgroup. **b** MetCaCb-like protein pairs formed hexamers. The method for co-purification of AcMetCaCb, HmMetCaCb, and RsMetCaCb was similar to that for MetCaCb. Notably, elution peaks at a volume of ~12.9 mL, indicative of hexameric protein complexes, were selected for subsequent enzymatic activity tests. Purification of HmMetCaCb was accompanied by an inactive peak (15.9 mL) which corresponded to the HmMetCa monomer. **c** Assessment of the metformin hydrolase activity of the MetCaCb-like protein pairs. The data represent the mean values of two replicates using independent enzyme preparation. Source data are provided as a Source Data file.

Thus, an additional member has been now added to the emerging group of pseudoenzyme-enzyme pairs with a characterized function.

Arginase family enzymes are commonly characterized as homomultimers[20]; however, their catalytic activity does not seem to depend on the multimeric state. For instance, disruption of the hexameric structure of arginase from *Bacillus caldovelox* by introducing an E278A variant exhibited similar kinetic parameters as those of the wild-type enzyme[34]. Additionally, eukaryotic arginases from humans and rats are naturally trimeric, whereas their monomers exhibited comparable activities as trimeric holoenzymes[35,36]. In this study, however, metformin hydrolase was identified as the first two-subunit arginase family enzyme composing a heterohexamer. Metformin hydrolase activity cannot be detected when mixing individually purified MetCa and MetCb in vitro (Fig. 1c), indicating the failure of forming MetCaCb complex. This could be due to the purified MetCb formed a well-organized homohexamer structure in solution, the interaction surfaces required for heteromultimeric association might be occupied by a preformed homomultimeric form of MetCb. The distinct interaction of MetCa with MetCb to form a protein complex was essential to the metformin hydrolase activity. This interaction enhanced the stability of MetCa in solution, and also contributed to both the substrate tunnel and fine-tuning of the active site.

The *metCa* and *metCb* genes were found as an adjacent gene pair in the same clade, as shown by the sequence similarity network analysis of the arginase family proteins (Supplementary Fig. 8). It is likely that *metCaCb* evolved via a gene duplication event, and the loss of catalytic activities of one of the proteins might have led it into a new evolutionary trajectory to become specialized as a protein binder. For the bacterial cell, the loss of the nickel-binding ability of the non-catalyzing MetCb subunit relieves the intracellular demand for the highly toxic nickel ion and also the energy consumption for nickel insertion. Notably, inactive cognates of enzymes are commonly found, but their functions are not well understood[37], as is the cases with the carboxyguanidine deiminase (CdgAB) in the DUF1989 protein family[38] and the 2,4-dinitroanisole demethylase (DnhAB) in the metallo-beta-lactamase (MBL) family[39]. Nonetheless, these cases imply that the presence of heterocomplexes within the protein families extend the catalytic diversity of enzymes by using a limited number of available gene products.

The available evidence clearly indicated that metformin functions as its prototype structure within the human body, although the

mechanisms underlying the effect of metformin on reducing blood glucose levels in humans are intricate and still not fully understood[6]. Metformin has been extensively synthesized and used by human to treat diabetes and subsequently has been found in the environment for only a few decades. Our results revealed that a set of arginase family enzymes found in the past few years from environmental bacteria are rapidly evolved to become active hydrolases against metformin. The metformin hydrolase is not a particularly efficient enzyme with a relative high $K_m$ value and low catalytic efficiency (Supplementary Table 1). So it seems that the time period since the introduction of metformin into the environment has been too short to evolve an efficient enzyme. Nonetheless, the emergence of metformin hydrolase raises concerns about whether it may potentially be present in gut microbes, resultantly leading to the breakdown of the active prototype structure of metformin. Although we haven't observed metformin hydrolase homolog in the human microbiome, it should be noticed that a case has been observed regarding acarbose, another antidiabetic drug, which is rendered inactive by enzymes present in the human gut and oral microbes[40,41]. Considering the significant secondary failure rate of metformin monotherapy in clinical practice[42], the potential presence of metformin hydrolase in the human gut microbiome should be carefully investigated due to its implications for this important medication.

The optimal catalytic conditions are ~50 °C and pH ~9.0 for metformin hydrolase MetCaCb (Fig. 2d, e). It is prevalent that arginase family enzymes prefer reaction conditions that are alkaline and around 50–60 °C, including human arginases and arginases from gut microbes[20]. It seems that these enzymes function under physiological conditions, although they probably never operate near their optimum conditions. The metformin hydrolase has been identified from bacteria found in wastewater from urban domestic sewage treatment plants[17], and we haven't observed its homolog in the human gut microbes as aforementioned. Currently, the enzyme most likely works in environments such as water, soil, etc, where the strain was isolated in this case.

## Methods

### Growth conditions of *Aminobacter* sp. strain NyZ550

*Aminobacter* sp. strain NyZ550, a metformin utilizer, was isolated as previously described[17]. The cells were cultivated in 1/5 strength tryptic soy broth (TSB) in Erlenmeyer flasks at 30 °C in a shaking incubator at 180 rpm. When testing the substrate-utilizing ability of the wild-type strain NyZ550 and its derivatives, the cells were cultivated in nitrogen-free minimal salts medium (MSM, pH 7.0)[43] supplemented with 5 mM metformin or dimethylamine. Ammonium sulfate (0.2%, w/v) was used as a nitrogen source if necessary. Biomass accumulation was estimated by measurement of the optical density at 600 nm ($OD_{600nm}$).

### Cloning, heterologous expression, and purification of recombinant proteins

The coding sequences of *metA*, *metB*, *metCa*, and *metCb* were amplified from genomic DNA of strain NyZ550 and subcloned into vector pUC19 containing a *lac* promoter. The primers used can be found in Supplementary Data 1. For recombinant protein expression and purification, pMAL-c2X containing a maltose-binding protein (MBP) tag and pET-28a(+) containing a 6×His tag were used. Different combinations of the genes were generated by overlap PCR amplification and inserted into desired vectors by Gibson assembly to generate expression constructs. The DNA sequences of MetCaCb homologs were synthesized by a commercial company and inserted into a pET-28a(+) vector. Recombinant proteins were produced in *E. coli* Rosetta(DE3) harboring the expression constructs. The *E. coli* Rosetta(DE3) cells were grown in lysogeny broth (LB) with appropriate antibiotics at 37 °C to an $OD_{600nm}$ of 0.6. Then IPTG was added to a final concentration of 0.1 mM, and cells were induced at

16 °C with shaking for 12 h. After harvesting the cells by centrifugation, they were suspended in an elution buffer containing 50 mM Tris-HCl (pH 8.0), 100 mM NaCl and 5% glycerol and lysed by ultrasonication. Cell lysates were centrifuged at 16,000 g for 40 min at 4 °C. The soluble protein fractions were used for enzymatic assays or purification by loading onto affinity columns. For the His-tagged proteins, the lysates were loaded onto a nickel-nitrilotriacetic acid column (GE Healthcare) with elution buffer containing 25 mM imidazole, and eluted in elution buffer supplemented with 250 mM imidazole. For the MBP-tagged proteins, the lysates were loaded onto a 1-mL dextrin column (GE Healthcare). After washing with elution buffer, proteins were eluted with 10 mM maltose in elution buffer. The proteins were then subjected to size exclusion chromatography with a Superdex 200 Increase 10/300 GL column (GE Healthcare) using elution buffer for column equilibration and elution. The resulting fractions were concentrated by ultrafiltration using a 10-kDa Amicon Ultra tube (Millipore) and stored at −80 °C for further use.

### Molecular weight determination

The recombinant MetCaCb protein solution underwent dialysis against a glycerol-free buffer containing 20 mM Tris-HCl (pH 8.0) and 200 mM NaCl. The molecular weight was determined in a protein concentration of 6.0 mg/mL by size exclusion chromatography (SEC) coupled with multi-angle light scattering (MALS). Bovine serum albumin (Thermo Fisher Scientific, Waltham, MA) with a molecular weight of 66 kDa was used to normalize the static light scattering detector.

### Metformin hydrolase activity assays

Standard metformin hydrolase activity assays were carried out in 500 μL of a reaction mixture containing 50 mM Tris-HCl buffer (pH 9.0), 0.2 mM $NiSO_4$, and 0.1–5 μg of the purified enzyme or 300 mg of crude bacterial extracts. The reaction was initiated by adding 10 mM metformin or $^{13}C^{15}N$-labeled metformin and incubated at 50 °C for 60 min. The reaction was terminated by the addition of equal volumes of acetonitrile, and centrifugated at 15,000 g for 20 min before further high-performance liquid chromatography-mass spectrum (HPLC-MS) analysis. Enzyme assays were performed at an appropriate enzyme concentration for the linear production of guanylurea during the incubation period. For the reaction conditions and kinetic analyzes, the metformin concentration, temperature, pH, buffer, and transition metal ions were modified as indicated in the respective figure legends. Guanylurea and dimethylamine release was measured by HPLC-MS and ion chromatography, respectively, using methods described previously[17].

### NMR spectroscopy

The reaction mixtures were prepared as standard metformin hydrolase assays with 10 mM $^{13}C^{15}N$-labeled metformin (Sigma-Aldrich). Then, 0.4 mL of each reaction mixture was transferred to NMR tubes to record the $^{13}C$-NMR spectra on a Quantum-I Plus 400 MHz spectrometer (Q.One Instruments Ltd., China).

### Genetic manipulation

Construction of markerless mutants in strain NyZ550 was performed using a two-step homologous recombination procedure. Briefly, a downstream fragment and an upstream fragment of the target gene were amplified from the genomic DNA of strain NyZ550, and the resulting fragments were assembled together by overlap PCR. The primers used can be found in Supplementary Data 1. The recombinant fragment was then ligated to *Eco*RI- and *Bam*HI-digested pEX18Gm, derived from a suicide vector pEX18Tc by replacing the tetracycline resistance gene cassette with a gentamicin resistance gene cassette. The resulting pEX18Gm derivatives were transformed into

diaminopimelic acid auxotroph *E. coli* WM3064, which was then conjugated to strain NyZ550. The transconjugants bearing suicide plasmids were subjected to double-crossovers on an TSB agar plate containing 10 μg/mL gentamicin (for the first recombination) or 15% sucrose (for the second recombination) at 30 °C. The in-frame deletions of the target genes were confirmed via PCR and sequencing. To complement the mutants, the genes and their respective 150 bp-length promoter sequences were cloned by PCR and ligated to *Eco*RI- and *Bam*HI-digested pEX18Gm. The resultant plasmids were transformed into the mutants as the aforementioned method.

### Metal analysis of metformin hydrolase by ICP-MS
MetCaCb is purified from the *E. coli* cells grown in nickel-rich (0.5 mM NiSO₄) LB media. Proteins were buffer-exchanged into 20 mM Tris-HCl by performing gel filtration chromatography with a Superdex 200 Increase 10/300 GL column. The proteins were concentrated to 10 mg/mL and analyzed by ICP-MS. One milliliter of the flow-through buffer was collected and used as a blank.

### Crystallization and structure determination
The coding sequences of MetCaCb were amplified from genomic DNA and cloned into a pET-28a(+) vector by Gibson assembly for coexpression of MetCaCb in *E. coli* cells grown in LB media. The MetCa subunit contained an N-terminal 6×histidine tag. Purified MetCaCb was stored in a buffer containing 20 mM Tris-HCl (pH 8.0), 100 mM NaCl, and 5% (v/v) glycerol. The sitting-drop vapor-diffusion method was used for crystallization of MetCaCb with equal volumes (0.8 μL each) of protein (16 mg/mL) and reservoir solutions at 20 °C. The crystal grew from only one liquor condition: 0.15 M calcium chloride dihydrate, 0.1 M MES buffer system (pH 6.2), 5% (v/v) glycerol and 15% (v/v) PEG smear broad. Attempts to get crystals of enzyme-substrate complex or nickel-loading enzyme were unsuccessful.

For data collection, the crystal was soaked in the reservoir solution supplemented with 20% (v/v) glycerol as a cryoprotectant, and subsequently vitrified by flash freezing in liquid nitrogen. X-ray diffraction data were collected at 100 K using a wavelength of 1.0 Å on the BL17U beamline at the Shanghai Synchrotron Radiation Facility. Diffraction data were processed using the XDS program package[44], and phasing was solved by molecular replacement in PHASER[45]. The initial structures of MetCa and MetCa were produced by Alphafold2[46] with pLDDT values > 90, and then employed as search models. Automated model building and refinement were performed in PHENIX[47]. The program COOT[48] was used for visualization of the electron density map, as well as for manual adjustments and refinements of the model. The model quality was assessed by MolProbity[49]. The metformin molecule was docked into MetCaCb using AutoDock Vina v1.1.2[50] and the top model with an AutoDock score of −4.7 is shown.

### Sequence comparison and phylogenetic analysis
To explore the co-occurrence of MetCa and MetCb homologs in bacteria, amino acid sequences of MetCa and MetCb were used as queries to search against the NCBI nr database in June 2023 using Cblaster (v1.3.18) with a threshold of >25% identity and >70% coverage. All sequences of the retrieved homologous protein pairs were extracted through the extract_clusters module and manually curated with a gap threshold of less than 2000 bp between genes. The resulting sequence pairs (Supplementary Data 2) were used to construct a phylogenetic tree by the maximum likelihood method with human arginase I (hARGI, NCBI accession number: NP_000036.2) as the outgroup.

Multiple sequence alignment of MetCaCb with characterized sequences was performed by MUSCLE[51]. The sequences used included guanidine hydrolase (GdmH), agmatinase (SpeB), guanidinobutyrase (GbuA), guanidinopropionase (GpuA), arginase from *Thermus thermophilus* (ttARG), human arginases I and II (hARGI and hARGII), and arginase from *Schistosoma mansoni* (smARG).

### CryoEM data acquisition and image processing
MetCaCb was purified following the aforementioned procedure, and then 3.5 μL of the sample was applied to Quantifoil 1.2/1.3 copper grids and mounted onto a Vitrobot Mark IV. Subsequently, the sample was blotted with a filter paper for a duration of 2 s using a blot force of −1. The grids were promptly immersed into liquid ethane to ensure their rapid cryopreservation.

Data acquisition was carried out at the Instrumental Analysis Center of the Shanghai Jiao Tong University with a Thermo Scientific Glacios transmission electron microscope operated at 200 kV and equipped with a Falcon 3 camera. Automated data collection was carried out using EPU software in linear mode. In total, 1068 movies were taken at a defocus range between −1.5 and −2.5 μm, calibrated pixel size of 1.6 Å/pixel, and a total dose of 40 electrons/Å² was applied to the movies over an exposure time of 2.54 s.

Image processing was performed by cryoSPARC (v3.1). For pre-processing, beam-induced motion and the contrast transfer function (CTF) was corrected on the movies using cryoSPARC's own implementations Patch Motion Correction and Patch CTF Estimation, respectively. Particles were selected from pre-processed micrographs using the blob picker. A total of 2,052,983 particles were picked and extracted from the micrographs. Subsequently, a subset of particles was selected after several rounds of 2D classification to build the initial model using Ab initio reconstruction. Then, several rounds of heterogeneous refinement ($n = 2$) were used to improve the initial map quality. The final map of 3D classification was further processed using non-uniform (NU) refinement and resulted in a map of approximately 5.0 Å resolution based on a Fourier Shell Correlation (FSC) cutoff at 0.143.

### Reporting summary
Further information on research design is available in the Nature Portfolio Reporting Summary linked to this article.

## Data availability
The crystal structure data of MetCaCb generated in this study have been deposited in the Protein Data Bank (PDB) database under accession code 8X3G. The cryoEM movies and the refined model are available from the corresponding author upon request. Source data are provided with this paper.

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

## Acknowledgements

This work was supported by the National Natural Science Foundation of China (NSFC) (32230001). We thank Kaixuan Gao for the assistance with the CryoEM data processing, and Prof. Yu-Zhong Zhang of Ocean University of China for stimulating discussion.

## Author contributions

T.L. and N.Y.Z developed the research plan and experimental design. Z.J.X. and T.L. performed the molecular and biochemical experiments. J.X. constructed the gene-knockout variants. T.L. and Z.J.X. isolated and crystallized proteins. T.L. and S.T.Z. collected and analyzed crystallographic data. P.P. performed the bioinformatic analysis. T.L., Z.J.X. and N.Y.Z wrote the manuscript, which was read and approved by all authors.

## Competing interests

The authors declare no competing interests.
