## [Peer Review File · Nature Communications]

Reviewers' Comments:

Reviewer #1:

Remarks to the Author:

In this manuscript, Li et al characterize a fascinating enzymatic system that enables bacterial catabolism of the common medication metformin via a metal-dependent hydrolase forming guanyurea and dimethylamine. This is a really beautiful paper and enzymatic system. Not only does MetCaCb present distinctive chemical reactivity, but also the unique evolution of two separate genes encoding arginase proteins (one catalytic, and one non-catalytic), both required for enzymatic catalysis. The evolution and mechanism of this metformin hydrolase could have great impact on understanding microbial catabolism in the ancient arginase hydrolase family, and potentially contribute to our understanding of metformin is a prominent drug, particularly in diabetes type 2 treatment.

Overall I recommend this manuscript for publication with minor revisions; these data present exciting implications for both metformin catabolism and for the wider arginase protein family.

Recommendations:

- One of the most intriguing parts of this paper, and point emphasized in this paper, is the impact that this catabolic pathway might have on gut microbial catabolism in human hosts who are administrated metformin. Authors use the Enzyme Function Initiative (EFI) Enzyme Similarity Tool (EST) to form a sequence similarity network (SSN) (see Extended Data 7). Using the EFI's Computationally-Guided Functional Profiling Tool (CGFP) to observe species that encode for these genes in the human microbiome would be highly valuable. EST-CGFP is a very user-friendly, publicly available resource that searches human metagenomic databases for prominence of these genes encoded in the human microbiome—running an analysis to see how abundant your genes of interest are in the human microbiome (particularly gut) would strengthen the impact of this paper. If these enzymes are abundant in human microbial species, this would be a compelling piece of data (possibly even to include in main figures).
- I noticed how optimal catalytic temperatures are $\sim 50^{\circ}\text{C}$ and pH ~ 9.0 . These are notably different than human body temp ($\sim 37^{\circ}\text{C}$ and closer to neutral, or even acidic pHs). What are thoughts of human biological relevance keeping in mind these experimental results?
- Do you still have crystals, and if so, could you collect anomalous data at the Ni K-edge (~ 8330 eV/1.49 angstrom)? This would strength argument of this enzymes specifically being Ni vs other metal (such as Mn dependent). What might be acompetitive advantage of using Ni vs other more abundant metals in the cell?
- It is so cool how MetCa and MetCb are so similar, both by sequence and overall architecture. It could be helpful to describe in text the residues that specifically dictate the 2:1 stoichiometry. Are there conserved binding residue pairs that enable this binding and the importance of the non-catalytic MetCb subunit? Is there any speculation as to why MetCa and MetCb would have evolved as this heterooligomer vs homoligomer? Authors suggest that this evolved via a gene duplication event—how conserved is this among species, and what might be a competitive advantage that would allow this heterooligomeric species to evolve, distinct from other members of the arginase enzymes.
- You discuss how MetCb proteins do not have residues to enable di-metal coordination. Are these mutations conserved across species, and are all those genes co-localized with a gene like MetCa in the genome? Also it would be helpful to show crystallographic composite omit maps to convince reader that metal is not bound in this extremely similar protein—this is an extremely cool system but I want you all to strength the argument of how/why MetCb might have evolved and whether it's conserved among other members of the family, possibly associated with other arginase enzymes.
- You all discuss the N- and C-terminal tails in MetCa and MetCb and how those might regulate oligomeric state. These regions look relatively flexible (at least in Fig 3b). What are the B-factors in these regions? Are the orientation similar in the crystal vs EM structures? Could those orientations in the crystal structures possibly be held in place by crystallographic contacts?
- Authors mention residues involved in substrate binding (for example S289, page 12, line 239). How do you know this residue is involved with binding? If only from computational docking, I

would make it clear how these are hypotheses. Is it conserved among species? Similarly authors state how residues such as N199 (similar to an equivalent His) are involved with proton shuffling. Be clear on how these conclusions are being drawn, and if the basis is solely on computational docking. Also a figure with a Chemdraw proposed mechanism would help readers interpret such statements. If authors are unable to get a substrate (or analog) bound structure, comparison to other members of the enzyme family and their proposed substrate binding and mechanisms would be helpful.

- Along with thinking about mechanism, how is Ni redox state reset and why do you think this enzyme might be Ni-specific from a mechanistic standpoint? Can you gain insight for hypotheses based on other members of the enzyme family?

Minor notes:

1) Double check for minor typos, including these that I noticed:

a. Page 4, Line 68: change to "produce guanylurea and dimethylamine"

b. Page 5, Line 84: you mention the MerR family, do you mean MerR? Just wondering because I only see MetR throughout paper

c. Page 5, Line 90: change to "usually preferring Mn²⁺ for the catalysis of cleavage of C-N bonds"

d. You all write the following line. "The hydrolase activity was inhibited by the arginase inhibitor 2(S)-amino-6-borohexonic acid (ABH). Indicating the possible formation of a tetrahedral intermediate during MetCaCb-catalyzed metformin hydrolysis. This sentence confused me without any shown mechanisms, including intermediates. Showing an hypothesized mechanism would help clarify the story and for visualization of statements such as this.

e. Page 19, Line: 389 change "1-ml" to 1-mL)

f. Page 19, Line 404: change "500 uLof" to "500 uL of"

2) Please include CC1/2 in X-ray data processing statistics table.

3) Please include occupancy values in X-ray refinement statistics table.

4) I was a little surprised by activity of MetCaCb with Mn in Figure 1b vs 4c with Mn and no metal. Can authors please clarify their thoughts, and also thoughts on Ni specificity, especially since this is so heavily emphasized in the paper and even title?

5) A cryo-EM data table would be useful. Most of this info is in methods but would be helpful to have in table like crystallography. Useful info would include: grid type, microscope/detector, voltage, recording mode, dose rate, defocus, pixel size, total dose, frame rate, number of frames/movie, exposure time, number of micrographs, number of micrographs used, total picked particles, number of used particles. A figure showing local resolution by color of protein complex would be helpful for observing which parts of structure are higher resolution than others.

Overall I loved reading this paper. It is very clear and a really fascinating enzyme system (especially as it requires two similar yet distinct subunits). Clarifications and additions discussed above would be helpful, but overall I am very excited for this work to be published and recommend it for publication upon minor revisions and additions.

Reviewer #2:

Remarks to the Author:

The authors describe the characterization of a heteromeric metforminase that is active as a hexamer composed as a centrosymmetric dimer of trimers. Interestingly, the genes encoding each subunit must be coexpressed in *E. coli* to produce active enzyme (no activity if genes are expressed individually and mixed). The authors show that the enzyme is highly specific for metformin and is Ni dependent. Knocking out the genes in *Aminobacter* prevented the bacterium from growing on metformin, confirming that the metforminase is necessary and sufficient. A phylogenetic analysis suggested that metforminase evolved from a ureohydrolase family enzyme, and the authors characterized representative enzymes for metforminase activity. X-ray crystallography and other techniques were used to determine the quaternary structure of the enzyme.

The manuscript is well-written, the figures are effective, and the experimental work seems to be very high quality. I have only a few minor suggestions:

1. The authors show that a metAB mutant also lost the ability to grow on metformin. The interpretation was that MetAB is required for metforminase activity in vivo, possibly by providing Ni to the enzyme where Ni levels are low. However, since metAB genes are directly upstream the metforminase genes in the genome, it is possible that deleting metAB interferes with expression of one or both metforminase genes, and this is the reason the mutant doesn't grow on metformin. I recommend the authors determine whether metforminase genes are being expressed in the metAB mutant by qPCR or some other method. Showing that they are expressed would strongly suggest that MetAB are indeed required in vivo and support their role in Ni loading.

2. In the discussion the authors state that "our results revealed the emergence of a set of metformin hydrolases in environmental bacteria that are rapidly evolving to become efficient." I believe this is true, but could any evidence be provided to back this up? For example, can the authors determine which year metforminase sequences started to appear in metasequence data? If sequences only appear in the past few years, that would seem to provide evidence that these genes recently evolved.

Reviewer #3:

Remarks to the Author:

The manuscript by Li et al. deals with the bacterial degradation of metformin, an anthropogenic biguanidine derivative. It is used therapeutically, particularly for the treatment of type 2 diabetes, and is one of the most frequently released active pharmaceutical compounds worldwide. Bacterial degradation by aquatic and soil bacteria is considered an important process in the removal of metformin from the environment. In recent years, metformin-degrading bacteria have been isolated and gene clusters and enzymes involved in metformin degradation have been identified. Only the first enzyme of the degradation pathway, which cleaves metformin into guanidylurea and dimethylamine, has not yet been studied in detail.

In the present manuscript by Li et al., metformin hydrolase was produced heterologously from an *Aminobacter* sp. for the first time and characterized kinetically, biochemically, structurally (X-ray crystallography and cryo-EM) and phylogenetically. The hydrolase from the arginase protein family is a di-nickel-center-containing heterohexameric enzyme ($\alpha_2\beta_4$), of which only the α -subunit carries a catalytic center. However, the enzyme is only active in the presence of both subunits. A phylogenetic analysis identified further potential metformin-cleaving enzymes, some of which were heterologously produced in active (albeit weak) form.

Overall, it is a very comprehensive, solid work in which the first enzyme of metformin degradation was proven beyond doubt. The significance of the work for the field of research stems from the widespread use and distribution of metformin. Although the hydrolase itself is unusual in its subunit composition, the catalyzed reaction is rather trivial and typical for members of the well-studied ureido-hydrolase family. Overall, the work is very well written, and also scientifically at a very high level.

Major points:

In particular, the figures and their captions are too small and hard to read, at least in the form I have. The authors should check the legibility of all illustrations (contrast of color, font size, overall size...).

In the discussion, one could elaborate on the aspect that metformin is a xenobiotic that was first introduced into the environment by humans. It is therefore not surprising that metformin hydrolase is not a particularly efficient enzyme (e.g. very high K_m value). Also those enzymes shown in Fig. 4c are, if at all, only extremely poor metformin hydrolases. So it seems that the time since the introduction of metformin has been somewhat very short to evolve an efficient enzyme.

Minor points:

- Fig. 2b: indicate in the legend the symbol for activity. The authors should use a common style of giving activities. In Fig 1a, a rate, in Fig. 2b, a final product concentration is given. At least, the authors should indicate after which incubation time the product was formed.
- Suppl. Fig 1ab: the authors should describe what is shown in the lanes that were cropped
- Ext. Fig. 2 and throughout the manuscript. Instead of using N.D., the authors should rather state that the concentration is under the detection limit, and give a number for the latter (i.e. <0.01 mM)
- L.752 replace 'with' a concentration by 'at' a concentration
- L.171: replace 'recognition' by 'specific binding'
- Fig. 3a. It is difficult to spot the metals (green spheres) in the printed version: too small. Fig 3c, for labelling of amino acids, use colours with more contrast. The amino acids in Fig. 3e can hardly be seen.
- L.180: ...was roughly inferred.. better: ... is reflected by the different..... or ...explains the different....
- Ext. Fig 4a: It should be zoomed in to better recognize the micrograph
- L.210 ff.: this is not clear to me: the enzyme as isolated had almost no Ni, but Mn, which explains the low activity (Fig. 1b)?? The authors may place metal content analysis better at the beginning, because it fits more to the reconstitution assays shown in Fig. 1b
- Comment on the low activity of heterologously produced enzymes, do you believe that they prefer other substrates

Reviewer #4:

Remarks to the Author:

In a previous paper by the authors, they showed that *Aminobacter* sp. strain NyZ550 (67-69) and an engineered *Pseudomonas putida* PaW340 expressing a guanylyurea hydrolase could together achieve the degradation of metformin. (lines 734-735) These results demonstrate the feasibility of a microbial degradation pathway for this widely prescribed anti-diabetic and anti-cancer drug. Understanding the molecular machinery that catalyzes this transformation is a worthwhile endeavor given the environmental burden associated with the presence of metformin in surface and wastewater, and its likely future expansion. The authors have reached a good level of understanding, as evidenced by the large amount of biophysical, experimental, and spectroscopic data that have been carefully compiled. However, in my opinion, the results presented here are only a first step in this understanding of the mechanism of action. There are still crucial questions that remain unanswered, such as the question of Ni vs. Mn, etc. Below are my comments and questions, the purpose of which is to have answers to some important questions that are still unclear at this time.

See below for my main concerns:

The authors carefully demonstrate that the MetCaCb enzyme catalyzes the hydrolysis of metformin in a Ni(II) ion-dependent manner (lines 151-153) - indeed, no other ion was found to support efficient catalysis (see Fig. 1b) (lines 665-669). However, according to the ICP-MS results (lines 210-213), the purified sample of the enzyme subsequently used for both crystallographic and cryo-EM studies contains Mn(II) instead of Ni(II).

More surprisingly, only a single Mn(II) ion appears to be present in the double-localized active site without the addition of excess Mn(II). The immediate environment of the metal coordinating site is homologous to that of arginases, which function by coordinating two Mn(II) ions. In addition, MetCaCb has only 2 His residues in the vicinity of the metal binding sites, whereas metallo-ureases catalyzing by coordination of 2 Ni(II) ions generally use at least 4 His residues to maintain secure coordination. The problem, therefore, is that the determined structures in which the Ni(ii) ion(s) is (are) missing may refer to an inactive form of the enzyme. Of course, it is possible that the Ni(II)-loaded form is isostructural with the Mn(II)-coordinating state, or that Ni(II) is bound to another, different - perhaps allosteric - site, but this has not yet been clarified. 1) It is not clear why the authors did not crystallize the enzyme from a solution containing Ni(II) as well, or if this is not possible, 2) at least soak the preformed crystals with Ni(II) ions. 3) Similarly, the addition of Ni(II) during the cryo-EM measurements could also have been a possibility. This is a rather exciting puzzle. It deserves a detailed analysis, without which the present results appear to be preliminary only.

- It is also quite intriguing that the active heterohexameric form ($\alpha_2\beta_4$) of the enzyme MetCaCb cannot be formed by mixing purified samples of MetCa and MetCb (lines 109-110). However, no explanation for this phenomenon is given in the manuscript. 1) Is it possible that the interaction surfaces required for hetero-multimeric association are covered by a preformed homomultimeric form of either MetCa or MetCb (or both)? 2) Or is there a conformational change that locks the MetCa and MetCb samples into unassociable forms? A model is to be constructed and proposed that can account for these results!

-

A 5.6 Å resolution cryo-EM map of the MetCaCb system was obtained, confirming that the heterohexameric topology seen in the crystal form is indeed the one present in the solution state. However, to show their similarities, the crystal structure should be fitted to the cryo-EM map and the differences characterized and interpreted.

- The authors discuss key structural features contributing to metformin catalysis based on a single AutoDock-generated docked pose (lines 229-241). In order to support an *in silico*

generated model describing the enzyme-substrate complex, the stability of the proposed arrangement needs to be demonstrated by the application of conformational analysis, MD simulation, or QM/MM calculations. This discussion, in its present form, is simply not strong enough to be convincing.

- The authors have mentioned that they did a docking of metformin to MetCa (line 229), but unfortunately the docking structure is not provided. Please provide it as a supplementary data file.
- Did the authors try to co-crystallize metformin with the inactivated versions of the MetCaCb enzyme (please clarify)?
- 2(S)-amino-6-borono-hexanoic acid (ABH) is an arginine mimetic molecule (having the same structure as arginine, but with the guanidino group replaced by a methyleneboronic acid group). It is claimed that ABH blocks the hydrolase activity of MetCaCb (lines 156-159). However, MetCaCb itself does not catalyze the hydrolysis of arginine (line 167). Thus, reasoning regarding the mechanism of metformin hydrolysis based on the behavior of the enzyme when adding ABH needs more support. Could the authors provide further evidence, either computational or experimental, to further support these claims?
- Asparagine is an unusual histidine-substituting proton-switching residue as presented here. The authors suggest that N199 in MetCa functions in this way (lines 237-240). It would be desirable to provide several examples from the literature where asparagine acts similarly?
- Would it be possible to study whether the N199H mutant maintains its catalytic activity?
- Lines 109-113 mention that no active enzyme complex is obtained by mixing and incubating individually expressed and purified MetCa and MetCb together. However, an active enzyme can be obtained when the two are co-expressed. This would imply that either the heteromultimer cannot form after mixing, or that it does form but assumes an incorrect fold. Have you tried to measure the affinity of the individually/separately expressed subunits to each other? Were you able to determine that the individually expressed MetCa and MetCb proteins have the same monomeric fold as in the complex?
- Please revise the sentence starting with “Comparable activity was detected...” (line 99), as it is hard to parse.
- Fig. 3 panel e would be improved if the labeled residues were highlighted with a different color and/or if the sticks were made thicker. In Extended Data Fig. 6, it is difficult to see both the stick representation of the structural elements and the residue labels, please modify.

Reviewer #1:

In this manuscript, Li et al characterize a fascinating enzymatic system that enables bacterial catabolism of the common medication metformin via a metal-dependent hydrolase forming guanylurea and dimethylamine. This is a really beautiful paper and enzymatic system. Not only does MetCaCb present distinctive chemical reactivity, but also the unique evolution of two separate genes encoding arginase proteins (one catalytic, and one non-catalytic), both required for enzymatic catalysis. The evolution and mechanism of this metformin hydrolase could have great impact on understanding microbial catabolism in the ancient arginase hydrolase family, and potentially contribute to our understanding of metformin is a prominent drug, particularly in diabetes type 2 treatment.

Overall I recommend this manuscript for publication with minor revisions; these data present exciting implications for both metformin catabolism and for the wider arginase protein family.

Recommendations:

(1) One of the most intriguing parts of this paper, and point emphasized in this paper, is the impact that this catabolic pathway might have on gut microbial catabolism in human hosts who are administrated metformin. Authors use the Enzyme Function Initiative (EFI) Enzyme Similarity Tool (EST) to form a sequence similarity network (SSN) (see Extended Data 7). Using the EFI's Computationally-Guided Functional Profiling Tool (CGFP) to observe species that encode for these genes in the human microbiome would be highly valuable. EST-CGFP is a very user-friendly, publicly available resource that searches human metagenomic databases for prominence of these genes encoded in the human microbiome—running an analysis to see how abundant your genes of interest are in the human microbiome (particularly gut) would strengthen the impact of this paper. If these enzymes are abundant in human microbial species, this would be a compelling piece of data (possibly even to include in main figures).

Thank you for your pertinent suggestion, and we have now performed such analysis using the EFI-CGFP tool. It turned out that no metformin hydrolase homolog was identified in over 380 metagenomes from the human microbiome database. Moreover, as our analysis stated in the first version (Fig. 4A), we also didn't observe any hits originating from human microbiome when searching for the clustered metformin hydrolase genes in the NCBI database. We have now included this information in the revised MS (lines 372-373). We will keep performing this search from time to time.

Although the metformin hydrolase seems not to be currently present in the human microbiome, the emerging of metforminase activity in bacteria concerns us as well, at least from the following aspects: i) the potential evolution of metforminase from a distinct genetic origin (i.e., isoenzyme) in specific environment and selective pressure in the human gut; ii) the possible dispersion of metforminase genes, such as the *metCaCb* identified here, from the environmental microbes to human gut microbiome.

(2) I noticed how optimal catalytic temperatures are ~50°C and pH ~9.0. These are notably different than human body temp (~37°C and closer to neutral, or even acidic

pHs). What are thoughts of human biological relevance keeping in mind these experimental results?

It is prevalent that arginase family enzymes prefer reaction conditions that are alkaline and around 50-60°C, including human arginases [1] and arginases from gut microbes [2]. It seems that these enzymes function under physiological conditions, although they probably never operate near their optimum conditions. The metformin hydrolase has been identified from bacteria found in wastewater from urban domestic sewage treatment plants, and we haven't observed its homolog in the human gut microbes as aforementioned. Therefore, this enzyme most likely works in environments such as water, soil, etc, where the strain was isolated in this case.

(3) Do you still have crystals, and if so, could you collect anomalous data at the Ni K-edge (~8330 eV/1.49 angstrom)? This would strength argument of this enzymes specifically being Ni vs other metal (such as Mn dependent). What might be a competitive advantage of using Ni vs other more abundant metals in the cell?

Thank you for your constructive suggestion. Although we no longer have crystals, we repeated the crystallization experiment during this revision and obtained new ones for collecting anomalous data at 8347eV. Unfortunately, the diffraction data are challenging for further analysis due to either weak signal or some kind of twin crystal. In order to determine the Ni content of the MetCaCb, we also repeated the ICP-MS analysis using the MetCaCb purified from the *E. coli* cells grown in nickel-rich medium rather than LB medium previously used, resulting in an observation of 3.9 nickel atoms per $\alpha_2\beta_4$ hexamer of MetCaCb, consistent with the di-nickel center. Under physiological conditions, the intracellular nickel concentration is low and the load of nickel onto MetCaCb is assisted by the nickel incorporation-associated proteins (MetA and MetB), as evidenced by the inability of the *metAB* knockout strain to grow with metformin (Fig. 1e). Taken together, these results further support the assertion that the metformin hydrolase is a di-nickel enzyme.

Most arginase family enzymes prefer manganese for their catalysis; however, the metformin and guanidine hydrolases [3] are the only two members in this family which are activated by nickel. Both substrates metformin and guanidine have a more resonance stabilized structure compared to the arginine, the substrate of arginases. Therefore, the nickel-dependency of metformin and guanidine hydrolases is likely due to the higher Lewis acidity of nickel than manganese, which is likely required for catalyzing the hydrolysis of metformin and guanidine. These discussions have been included in the revised MS (lines 297-301).

(4) It is so cool how MetCa and MetCb are so similar, both by sequence and overall architecture. It could be helpful to describe in text the residues that specifically dictate the 2:1 stoichiometry. Are there conserved binding residue pairs that enable this binding and the importance of the non-catalytic MetCb subunit?

Thank you for the constructive suggestions that led us to ruminate deeper into the interaction and evolutionary scenario of MetCa and MetCb. As suggested, the interacting residues of MetCa and MetCb were analyzed. While the residues dictating

the binding of heterologous multimer seem currently enigmatic, a large number of mutation analyses would be required to determine if certain residues influence the heterologous binding. So far, we can propose that the N-terminal tail of MetCb, located at the interface of three subunits ($\alpha 1\beta 1\beta 4$ or $\alpha 2\beta 2\beta 3$) (Supplementary Fig. 5b), is crucial for the heterologous binding. Notably, truncating the 23 amino acids in the N-terminal of MetCb resulted in a failure to achieve soluble expression of the MetCaCb complex, suggesting its contribution to the formation of the heteromultimeric protein complex. These analyses have been strengthened in the MS (lines 230-233).

Is there any speculation as to why MetCa and MetCb would have evolved as this heterooligomer vs homooligomer? Authors suggest that this evolved via a gene duplication event—how conserved is this among species, and what might be a competitive advantage that would allow this heterooligomeric species to evolve, distinct from other members of the arginase enzymes.

Based on the sequences, structures, and gene organization of MetCa and MetCb, we hypothesized that MetCaCb evolved from a common ancestor, likely encoding a homomeric protein, through a gene duplication event and subsequent divergences, resulting in heteromeric complexes. The evolution of the heterooligomer appears to precede the emergence of metformin hydrolase activity, as evidenced by the lack of activity or extremely low activity of the heterohexameric homologs of MetCaCb (Fig. 4). Efficient metformin hydrolases have only been identified in *Aminobacter* spp. and *Pseudomonas* spp., indicating that the metformin hydrolase offers distinct advantages specific to these microorganisms.

Generally, the duplication of a gene encoding for an oligomeric protein can lead to numerous possible outcomes, and the changes of oligomeric states of descendant homologs are always associated with functional innovations. Compared to the homomeric state, the evolution of heterooligomeric complexes might confer functional versatility by allowing for more intricate regulation and fine-tuning of catalytic efficiency or substrate specificity, potentially enhancing the organism's ability to adapt to diverse metabolic demands or environmental challenges. For homomeric enzymes with catalytic sites present in each of subunits, such as arginase family enzymes, the scope of evolving non-catalytic regulation is relatively limited because each subunit is constrained to retain the primary function. However, for heteromeric enzymes, since two or more proteins are mediating the primary function, selection pressure may relax and allow the emergence of secondary functions such as structural scaffold. This opens numerous possible routes of functional innovation. Although heteromers are less common in prokaryotic proteomes than eukaryotic proteomes [4], the MetCaCb would be a sophisticated example of heteromeric enzyme in bacteria, and the story of the MetCaCb evolution would be of great interest in further investigation.

(5) You discuss how MetCb proteins do not have residues to enable di-metal coordination. Are these mutations conserved across species, and are all those genes co-localized with a gene like MetCa in the genome? Also it would be helpful to show crystallographic composite omit maps to convince reader that metal is not bound in this

extremely similar protein—this is an extremely cool system but I want you all to strength the argument of how/why MetCb might have evolved and whether it's conserved among other members of the family, possibly associated with other arginase enzymes.

We analyzed the MetCb homologs containing mutations in di-metal coordinating residues (Fig. 4). These proteins exhibit AA sequence identities of 58%-100% (coverage >97%) to MetCb. All of these protein-encoding genes are co-localized with a MetCa-like gene and are distributed in the phyla of bacteria including the *Proteobacteria*, *Actinomycetota* and *Acidobacteriota* (Fig. 4). The MetCb-like proteins from *Aminobacter* and *Pseudomonas* contain five identical mutations in di-metal coordinating residues to MetCb, while three or four mutations present in others (Fig. 4). The crystallographic composite omit maps have been included in the modified Supplementary Fig. 5c, d, clearly indicating that no metal ion is bound in MetCb.

(6) You all discuss the N- and C-terminal tails in MetCa and MetCb and how those might regulate oligomeric state. These regions look relatively flexible (at least in Fig 3b). What are the B-factors in these regions? Are the orientation similar in the crystal vs EM structures? Could those orientations in the crystal structures possibly be held in place by crystallographic contacts?

The average B-factor of the discussed N- and C-terminal tails in MetCa and MetCb has been indicated in the text (lines 198 and 230) and colored in the Supplementary Fig. 5b. The N-terminal tails of MetCa are stable which contributes to the interaction of the two MetCa subunits. The four MetCb subunits adopt two types of conformation, i.e., $\beta 1\beta 3$ in one conformation and $\beta 2\beta 4$ in another (Fig. 3a), and the N- and C-terminal tails of them seem to exhibit different flexibility. The electron density of N- and C-terminal tails can only be seen in the $\beta 1\beta 3$ subunits, whereas they are missing in the $\beta 2\beta 4$ subunits. This could be due to that the interaction of N- and C-terminal tails of $\beta 1\beta 3$ with the MetCa subunits stabilized their conformation. Although we are unable to distinguish the orientation in the CryoEM map due to the limitation of resolution, and cannot rule out the possibility that the crystallographic contacts contribute to the observed conformation of the N- and C-terminal tails in MetCa and MetCb, it should be noted that: 1) the N- and C-terminal tails of the $\beta 1\beta 3$ subunits exhibit significant polar contacts with MetCa (Supplementary Fig. 5b); 2) Truncation of these tails results in a failure to achieve soluble expression of the MetCaCb complex. Therefore, it is most likely that the interaction of the flexible tails of MetCb with MetCa exists in solution state and plays an important role in the heterohexameric architecture.

(7) Authors mention residues involved in substrate binding (for example S289, page 12, line 239). How do you know this residue is involved with binding? If only from computational docking, I would make it clear how these are hypotheses. Is it conserved among species? Similarly authors state how residues such as N199 (similar to an equivalent His) are involved with proton shuffling. Be clear on how these conclusions are being drawn, and if the basis is solely on computational docking. Also a figure with a Chemdraw proposed mechanism would help readers interpret such statements. If

authors are unable to get a substrate (or analog) bound structure, comparison to other members of the enzyme family and their proposed substrate binding and mechanisms would be helpful.

We agree that the currently evidences are too preliminary to elucidate the catalytic and binding details of this enzyme, and the statements have been rewritten: “The equivalent residue of N199 in the arginase family enzymes is a conserved noncoordinating histidine which is catalytically important during the reaction process. The activity of the N199H variant was significantly decreased, and the activity of the N199A variant is completely lost (Supplementary Table 1). The residue N199 likely plays a role similar to its equivalent histidine in arginases which is used for stabilizing the reaction intermediate. Additionally, the substrate pocket residue S289 and E320 was also essential for metformin hydrolase activity (Supplementary Table 1).”

(8) Along with thinking about mechanism, how is Ni redox state reset and why do you think this enzyme might be Ni-specific from a mechanistic standpoint? Can you gain insight for hypotheses based on other members of the enzyme family?

The nickel enzymes include redox enzymes and nonredox hydrolases. For the Ni redox enzymes, the Ni ion exhibits versatility in redox properties, exploiting the +3/+2 states (superoxide dismutase and hydrogenase), the +2/+1 states (hydrogenase, CO dehydrogenase, and acetyl-CoA synthase), and perhaps even reaching the 0 state (acetyl-CoA synthase) for the redox enzymes. On the other hand, the nonredox Ni hydrolases (e.g., urease) remain in the +2 state during catalysis. The metformin hydrolase apparently belongs to the latter, which appears to only use Ni²⁺ as a Lewis acid. Most arginase family enzymes prefer manganese for their catalysis; however, metformin and guanidine hydrolases [3] are the only two members in this family activated by nickel. Both metformin and guanidine have a more resonance-stabilized structure compared to arginine, the substrate of arginase. Therefore, the nickel-dependency of metformin and guanidine hydrolases would be due to that the Lewis acidity of nickel is higher than manganese, likely required for catalyzing the hydrolysis of metformin and guanidine. These discussions have been included in the revised MS (lines 296-300).

Minor notes:

(9) Double check for minor typos, including these that I noticed:

a. Page 4, Line 68: change to “produce guanylurea and dimethylamine”

Corrected

b. Page 5, Line 84: you mention the MerR family, do you mean MerR? Just wondering because I only see MetR throughout paper

Yes, it means MerR. For clarity, the “MerR family” has been revised to “MerR family regulator”.

c. Page 5, Line 90: change to “usually preferring Mn²⁺ for the catalysis of cleavage of C-N bonds”

Corrected

d. You all write the following line. “The hydrolase activity was inhibited by the arginase inhibitor 2(S)-amino-6-borohexonic acid (ABH). Indicating the possible formation of a tetrahedral intermediate during MetCaCb-catalyzed metformin hydrolysis. This sentence confused me without any shown mechanisms, including intermediates. Showing an hypothesized mechanism would help clarify the story and for visualization of statements such as this.

The statement “indicating the possible formation of a tetrahedral intermediate during MetCaCb-catalyzed metformin hydrolysis” has been removed.

e. Page 19, Line: 389 change “1-ml” to 1-mL)

Corrected

f. Page 19, Line 404: change “500 uLof” to “500 uL of”

Corrected

(10) Please include CC1/2 in X-ray data processing statistics table.

We have added it accordingly. Supplementary Table 2

(11) Please include occupancy values in X-ray refinement statistics table.

We have added it accordingly. Supplementary Table 2

(12) I was a little surprised by activity of MetCaCb with Mn in Figure 1b vs 4c with Mn and no metal. Can authors please clarify their thoughts, and also thoughts on Ni specificity, especially since this is so heavily emphasized in the paper and even title?

In Fig. 1b and Fig. 4c, we tested the activities of metformin hydrolase and its homologs, and we observed very low activity when no extra metal ion or manganese was added to the reaction mixtures. This could be due to the influence of a trace of nickel in the LB media used to grow the cells. Because analysis of the metal ion content of recombinant MetCaCb purifying from the LB-grown cells indicated that MetCaCb contains 0.8 manganese and 0.05 nickel atoms per $\alpha 1\beta 2$ trimer. Addition of manganese into the reaction mixture didn't evidently increase the activity of MetCaCb, due to the enzyme needs nickel to activate the reaction, as outlined in the response to points 3 and 8.

(13) A cryo-EM data table would be useful. Most of this info is in methods but would be helpful to have in table like crystallography. Useful info would include: grid type, microscope/detector, voltage, recording mode, dose rate, defocus, pixel size, total dose, frame rate, number of frames or movie, exposure time, number of micrographs, number of micrographs used, total picked particles, number of used particles. A figure showing local resolution by color of protein complex would be helpful for observing which parts of structure are higher resolution than others.

Thank you for the suggestions. A cryo-EM data table has been added as Supplementary

Table 3.

Overall I loved reading this paper. It is very clear and a really fascinating enzyme system (especially as it requires two similar yet distinct subunits). Clarifications and additions discussed above would be helpful, but overall I am very excited for this work to be published and recommend it for publication upon minor revisions and additions.

Reviewer #2 (Remarks to the Author):

The authors describe the characterization of a heteromeric metforminase that is active as a hexamer composed as a centrosymmetric dimer of trimers. Interestingly, the genes encoding each subunit must be coexpressed in *E. coli* to produce active enzyme (no activity if genes are expressed individually and mixed). The authors show that the enzyme is highly specific for metformin and is Ni dependent. Knocking out the genes in *Aminobacter* prevented the bacterium from growing on metformin, confirming that the metforminase is necessary and sufficient. A phylogenetic analysis suggested that metforminase evolved from a ureohydrolase family enzyme, and the authors characterized representative enzymes for metforminase activity. X-ray crystallography and other techniques were used to determine the quaternary structure of the enzyme.

The manuscript is well-written, the figures are effective, and the experimental work seems to be very high quality. I have only a few minor suggestions:

(1) The authors show that a *metAB* mutant also lost the ability to grow on metformin. The interpretation was that *MetAB* is required for metforminase activity *in vivo*, possibly by providing Ni to the enzyme where Ni levels are low. However, since *metAB* genes are directly upstream the metforminase genes in the genome, it is possible that deleting *metAB* interferes with expression of one or both metforminase genes, and this is the reason the mutant doesn't grow on metformin. I recommend the authors determine whether metforminase genes are being expressed in the *metAB* mutant by qPCR or some other method. Showing that they are expressed would strongly suggest that *MetAB* are indeed required *in vivo* and support their role in Ni loading.

We agree that the deletion of *metAB* might affect the expression of *metCaCb*, thereby resulting in its inability to grow on metformin. During the revision, we performed a gene complementation assay for the mutants. The results clearly indicate that the introduction of a plasmid expressing *metAB* into the *metAB* mutant recovered its ability to grow on metformin. This observation indicates the successful expression of *metCaCb* in the *metAB* mutant. We have included the results in the revised MS (Fig. 1e).

(2) In the discussion the authors state that “our results revealed the emergence of a set of metformin hydrolases in environmental bacteria that are rapidly evolving to become efficient.” I believe this is true, but could any evidence be provided to back this up? For example, can the authors determine which year metforminase sequences started to appear in metasequence data? If sequences only appear in the past few years, that would seem to provide evidence that these genes recently evolved.

The sequences from the clade of metforminases have appeared in the database since 2022. Therefore, the sentence has been revised to “our results revealed that a set of metformin hydrolases found in the past few years from environmental bacteria are rapidly evolving to become efficient.” (lines 364-366)

Reviewer #3 (Remarks to the Author):

The manuscript by Li et al. deals with the bacterial degradation of metformin, an anthropogenic biguanidine derivative. It is used therapeutically, particularly for the treatment of type 2 diabetes, and is one of the most frequently released active pharmaceutical compounds worldwide. Bacterial degradation by aquatic and soil bacteria is considered an important process in the removal of metformin from the environment. In recent years, metformin-degrading bacteria have been isolated and gene clusters and enzymes involved in metformin degradation have been identified. Only the first enzyme of the degradation pathway, which cleaves metformin into guanylurea and dimethylamine, has not yet been studied in detail.

In the present manuscript by Li et al., metformin hydrolase was produced heterologously from an *Aminobacter* sp. for the first time and characterized kinetically, biochemically, structurally (X-ray crystallography and cryo-EM) and phylogenetically. The hydrolase from the arginase protein family is a di-nickel-center-containing heterohexameric enzyme ($\alpha_2\beta_4$), of which only the α -subunit carries a catalytic center. However, the enzyme is only active in the presence of both subunits. A phylogenetic analysis identified further potential metformin-cleaving enzymes, some of which were heterologously produced in active (albeit weak) form.

Overall, it is a very comprehensive, solid work in which the first enzyme of metformin degradation was proven beyond doubt. The significance of the work for the field of research stems from the widespread use and distribution of metformin. Although the hydrolase itself is unusual in its subunit composition, the catalyzed reaction is rather trivial and typical for members of the well-studied ureido-hydrolase family. Overall, the work is very well written, and also scientifically at a very high level.

Major points:

(1) In particular, the figures and their captions are too small and hard to read, at least in the form I have. The authors should check the legibility of all illustrations (contrast of color, font size, overall size...).

We apologize for this inconvenience. We have modified the figures and their captions throughout in order to achieve more legibility.

(2) In the discussion, one could elaborate on the aspect that metformin is a xenobiotic that was first introduced into the environment by humans. It is therefore not surprising that metformin hydrolase is not a particularly efficient enzyme (e.g. very high K_m value). Also those enzymes shown in Fig. 4c are, if at all, only extremely poor metformin hydrolases. So it seems that the time since the introduction of metformin has been somewhat very short to evolve an efficient enzyme.

We agree and extend the discussion for including this point in the revised MS. It has now been stated in the discussion: “Metformin has been extensively synthesized and used by human to treat diabetes and subsequently has been found in the environment for only a few decades. Our results revealed that a set of arginase family enzymes found in the past few years from environmental bacteria are rapidly evolved to become active hydrolases against metformin. The metformin hydrolase is not a particularly efficient enzyme with a relative high K_m value and low catalytic efficiency (Supplementary Table 1). So it seems that the time since the introduction of metformin into the environment has been too short to evolve an efficient enzyme.” Lines 362-369

Minor points:

(1) Fig. 2b: indicate in the legend the symbol for activity. The authors should use a common style of giving activities. In Fig 1a, a rate, in Fig. 2b, a final product concentration is given. At least, the authors should indicate after which incubation time the product was formed.

The activities are indicated by reaction rate throughout except the Fig.2b. In Fig.2b, we aimed to determine the active fractions from SEC rather than to quantify the rate of the enzymatic reaction. Therefore, the activities of each fraction are indicated by the production of guanylyurea. We have now indicated the reaction conditions in the figure legend for Fig. 2b.

(2) Suppl. Fig 1ab: the authors should describe what is shown in the lanes that were cropped

Descriptions for all lanes in the uncropped SDS-PAGE images have been added in the legends, now it is in supplementary Fig. 8 and 9.

(3) Ext. Fig. 2 and throughout the manuscript. Instead of using N.D., the authors should rather state that the concentration is under the detection limit, and give a number for the latter (i.e. <0.01 mM)

We have given a number for the detection limit accordingly in the figure legends.

(4) L.752 replace ‘with’ a concentration by ‘at’ a concentration

Corrected. Supplementary Figure 2 legend.

(5) L.171: replace ‘recognition’ by ‘specific binding’

Corrected. Line 169.

(6) Fig. 3a. It is difficult to spot the metals (green spheres) in the printed version: too small. Fig 3c, for labelling of amino acids, use colours with more contrast. The amino acids in Fig. 3e can hardly been seen.

The figures have been improved as suggested.

(7) L.180: ...was roughly inferred.. better: ... is reflected by the different.... or ...explains the different....

Modified. Line 178.

(8) Ext. Fig 4a: It should be zoomed in to better recognize the micrograph
For better clarity, micrographs with different magnifications have been provided in the revised Supplementary Figure 4a.

(9) L.210 ff.: this is not clear to me: the enzyme as isolated had almost no Ni, but Mn, which explains the low activity (Fig. 1b)?? The authors may place metal content analysis better at the beginning, because it fits more to the reconstitution assays shown in Fig. 1b

In Fig. 1b, we tested the activities of metformin hydrolase with crude enzymes, and we observed very low activity when no extra metal ion or manganese was added to the reaction mixtures. This could be due to the influence of a trace of nickel in the LB media used to grow the cells. Because analysis of the metal ion content of recombinant MetCaCb purifying from the LB-grown cells indicated that MetCaCb contains 0.8 manganese and 0.05 nickel atoms per $\alpha 1\beta 2$ trimer. Addition of manganese into the reaction mixture can't significantly increase the activity of MetCaCb, as the enzyme needs nickel to activate the reaction, as outlined in the response to points 3 and 8 by reviewer 1.

In our original analysis, the enzyme was purified from LB media-grown *E. coli* cells without the addition of any extra metal ion, resulting in the observation of 0.8 manganese and 0.05 nickel atoms per $\alpha 1\beta 2$ trimer. We noted that this could be due to the limited content of nickel in the LB media. During revision, we performed an ICP-MS analysis for the recombinant MetCaCb purified from cells grown with nickel-rich LB media, and 1.95 nickel atoms per $\alpha 1\beta 2$ trimer were detected. We have now revised the MS accordingly. (lines 214-218)

(10) Comment on the low activity of heterologously produced enzymes, do you believe that they prefer other substrates

We agree that these low-activity homologous enzymes may well prefer other substrates; for example, the derivatives of guanidine or biguanide. Here, we focus on the characterization of a metformin hydrolase. However, in the future, it will be intriguing to investigate the substrate preference of metformin hydrolase and its homologs, as well as their evolutionary connections.

Reviewer #4 (Remarks to the Author):

In a previous paper by the authors, they showed that *Aminobacter* sp. strain NyZ550 (67-69) and an engineered *Pseudomonas putida* PaW340 expressing a guanylurea hydrolase could together achieve the degradation of metformin. (lines 734-735) These results demonstrate the feasibility of a microbial degradation pathway for this widely prescribed anti-diabetic and anti-cancer drug. Understanding the molecular machinery that catalyzes this transformation is a worthwhile endeavor given the environmental burden associated with the presence of metformin in surface and wastewater, and its likely future expansion. The authors have reached a good level of

understanding, as evidenced by the large amount of biophysical, experimental, and spectroscopic data that have been carefully compiled. However, in my opinion, the results presented here are only a first step in this understanding of the mechanism of action. There are still crucial questions that remain unanswered, such as the question of Ni vs. Mn, etc. Below are my comments and questions, the purpose of which is to have answers to some important questions that are still unclear at this time.

See below for my main concerns:

(1) The authors carefully demonstrate that the MetCaCb enzyme catalyzes the hydrolysis of metformin in a Ni(II) ion-dependent manner (lines 151-153) - indeed, no other ion was found to support efficient catalysis (see Fig. 1b) (lines 665-669). However, according to the ICP-MS results (lines 210-213), the purified sample of the enzyme subsequently used for both crystallographic and cryo-EM studies contains Mn(II) instead of Ni(II). More surprisingly, only a single Mn(II) ion appears to be present in the double-localized active site without the addition of excess Mn(II). The immediate environment of the metal coordinating site is homologous to that of arginases, which function by coordinating two Mn(II) ions. In addition, MetCaCb has only 2 His residues in the vicinity of the metal binding sites, whereas metallo-ureases catalyzing by coordination of 2 Ni(II) ions generally use at least 4 His residues to maintain secure coordination. The problem, therefore, is that the determined structures in which the Ni(ii) ion(s) is (are) missing may refer to an inactive form of the enzyme. Of course, it is possible that the Ni(II)-loaded form is isostructural with the Mn(II)-coordinating state, or that Ni(II) is bound to another, different - perhaps allosteric - site, but this has not yet been clarified. 1) It is not clear why the authors did not crystallize the enzyme from a solution containing Ni(II) as well, or if this is not possible, 2) at least soak the preformed crystals with Ni(II) ions. 3) Similarly, the addition of Ni(II) during the cryo-EM measurements could also have been a possibility. This is a rather exciting puzzle. It deserves a detailed analysis, without which the present results appear to be preliminary only.

In our initial metal ion content analysis of MetCaCb, the enzyme was purified from *E. coli* cells grown in LB media, the results indicated that only a single Mn(II) ion appears to be present in the active site. This could be due to the insufficiency of metal ion for completely loading or the loss of metal ion during protein purification [5]. For arginase family enzymes, the two metal ions (usually Mn²⁺) in the bimetallic center are not equally bound, and one metal ion is more loosely bound and more easily lost or removed than another one [6]. When incubation MetCaCb with Mn²⁺ at room temperature, we observed 1.9 manganese and 0.06 nickel atoms per trimer, indicating an intact bimetallic center is present in MetCaCb. However, the observed content of nickel atom is low. We noted that the nickel-dependent urease activity in a nickel insertion inability mutant could be restored with nickel supplementation in the growth medium [7]. We thus speculated that the nickel concentration in the growth medium may be insufficient for the competitively loading of nickel atom into overexpressed MetCaCb in *E. coli*. Therefore, we repeated the ICP-MS analysis using the MetCaCb purified from the *E. coli* cells grown in nickel-rich media, resulting in an observation

of 3.9 nickel atoms per $\alpha_2\beta_4$ hexamer of MetCaCb, consistent with a di-nickel center. We agree that to get nickel-loading crystal structure will be helpful, but our attempts regrettably failed. During revision, we repeated the crystallization experiment using newly purified nickel-loaded MetCaCb to get some crystals for diffraction, and also hope collecting anomalous data at Ni K edge at 8347eV. Unfortunately, the diffraction data are challenging for further analysis due to either weak signal or some kind of twin crystal. To test the possible structural change of Ni(II) or Mn(II)-coordinating states of MetCaCb, we performed a circular dichroism (CD) spectroscopy assay using Ni(II) or Mn(II)-loading enzymes. The CD results showed that the secondary structures of the enzymes are similar (see below), suggesting little structural alteration. Notably, the bi-Mn²⁺ center of arginase family enzymes adopts octahedral metal coordination, which is easy to be substituted with alternative metal ions that readily adopt octahedral metal coordination, such as Co²⁺ and Ni²⁺. The activity differences among different metal ion-substituted arginase may result from the differing abilities of each metal ion in activating the reaction through a metal-bridging hydroxide ion [8]. In addition, comparison of the structure of MetCaCb with the nickel-loaded structure of the nickel-dependent guanidine hydrolase [3] from this family indicated that they have an identical bimetallic center coordinating by two His and four Asp, differing from the di-nickel center of urease. However, the mechanism of nickel dependency of metformin hydrolase needs further careful investigation; similarly, the exact mechanism of the dependence of guanidine hydrolase on Ni²⁺ also needs to be elucidated.

Figure legend: CD spectra of Ni(II)-loading metformin hydrolase (red line) and Mn(II)-loading metformin hydrolase (blue line).

(2) It is also quite intriguing that the active heterohexameric form ($\alpha_2\beta_4$) of the enzyme MetCaCb cannot be formed by mixing purified samples of MetCa and MetCb (lines 109-110). However, no explanation for this phenomenon is given in the manuscript. 1) Is it possible that the interaction surfaces required for hetero-multimeric association are covered by a preformed homomultimeric form of either MetCa or MetCb (or both)? 2) Or is there a conformational change that locks the MetCa and MetCb samples into unassociable forms? A model is to be constructed and proposed that can account for these results!

This phenomenon is also observed for many multi-subunit heteromeric enzymes, with respect to MetCaCb, we observed that the purified MetCb formed a stable homohexamer state as evidenced by SEC analysis. This could prevent the formation of

heteromeric MetCaCb complex. We have now stated in the discussion: “Metformin hydrolase activity cannot be detected when mixing purified MetCa and MetCb in vitro (Fig. 1c), indicating the failure of forming MetCaCb complex. This could be due to the purified MetCb formed a well-organized homohexamer structure in solution, the interaction surfaces required for hetero-multimeric association might be occupied by a preformed homomultimeric form of MetCb.” Lines 336-341

(3) A 5.6 Å resolution cryo-EM map of the MetCaCb system was obtained, confirming that the heterohexameric topology seen in the crystal form is indeed the one present in the solution state. However, to show their similarities, the crystal structure should be fitted to the cryoEM map and the differences characterized and interpreted.

In the original Supplementary Fig. 4c, the crystal structure model of MetCaCb was overlaid onto the cryoEM map. To enhance clarity, we have refined the presentation images and amended the corresponding legend in the revised version. Also, features of the cryoEM map have been interpreted in the text: “To observe the polymeric state of the MetCaCb complex approximating physiological states, we performed single-particle cryoEM using the same sample for crystallization. Our analysis revealed a remarkable level of uniformity among the enzyme particles (Supplementary Fig. 4a, b). The resulting map presents as an asymmetric cylinder, with dimensions approximately 45 Å in radius and 80 Å in height. One side of the map exhibits tight cohesion, while the opposing side diverges to create an aperture. Subsequent fitting of the crystal structure model of the MetCaCb hexamer onto the cryoEM map provides unambiguous insights of the distribution of subunits within the enzyme: the tightly connected side corresponds to the interface where MetCa interacts, whereas the divergent region accommodates MetCb (Supplementary Fig. 4c). These findings adequately elucidate the hexameric configuration of the MetCaCb complex.”

(4) The authors discuss key structural features contributing to metformin catalysis based on a single AutoDock-generated docked pose (lines 229-241). In order to support an in silico generated model describing the enzyme-substrate complex, the stability of the proposed arrangement needs to be demonstrated by the application of conformational analysis, MD simulation, or QM/MM calculations. This discussion, in its present form, is simply not strong enough to be convincing.

The catalytic mechanism of di-metal dependent arginase family enzymes has been well established by studies on arginases from human, rat or bacteria [9, 10]. This family of enzymes shares a conserved catalytic mechanism by the conserved di-metal active center [9], as outlined by reviewer 3 above. Recently, the identification of the first nickel-dependent guanidine hydrolase from this family and elucidation of its crystal structure have provided us more details regarding nickel dependency of arginase family enzymes [3]. Essentially, the nickel-dependent guanidine hydrolase exhibits a similar catalytic mechanism to other arginase family enzymes. Here, by comparing the sequences and structures of the metformin hydrolase with the aforementioned arginase family enzymes, we discussed the properties of the residues in the substrate pocket of

MetCaCb. Molecular docking analysis was used to show that a metformin molecule can be successfully accommodated in the substrate pocket. Unfortunately, we did not obtain the enzyme-substrate complex structure to illustrate the specific binding details and to further perform molecular dynamic simulations for catalytic details; therefore, these details have not been claimed in the current version. In fact, we tentatively proposed the potentially key residues in the active site largely based on the comparisons with the well-known arginases (Supplementary Fig. 6b, c). It should be noted that we generated the metformin hydrolase variants by mutagenesis and performed enzymatic kinetics analyses, which support the important roles of these residues. Now, the corresponding content has been modified to more carefully present the results (lines 234-247).

(5) The authors have mentioned that they did a docking of metformin to MetCa (line 229), but unfortunately the docking structure is not provided. Please provide it as a supplementary data file.

Sorry for the inconvenience, the structure with docked ligand has been uploaded in the system. (file name: MetCaCb_metformin.PDB)

(6) Did the authors try to co-crystallize metformin with the inactivated versions of the MetCaCb enzyme (please clarify)?

We attempted to co-crystallize metformin with the inactivated variants of MetCaCb, including D183A, D187A, E320A and N199A. For each variant, we screened 480 crystallization conditions using commercial kits, and various metformin concentrations. Although we got a few needlelike crystals, sadly, none were suitable for X-ray diffraction. We have now clarified in the Method section (lines 474-475). Currently, efforts are being made to optimize the crystallization conditions for better crystals. Considering the significant challenges for obtaining useful structures of substrate-enzyme complexes of arginase family enzymes, this part has not been included in the present MS.

(7) 2(S)-amino-6-boronoheptanoic acid (ABH) is an arginine mimetic molecule (having the same structure as arginine, but with the guanidino group replaced by a methyleneboronic acid group). It is claimed that ABH blocks the hydrolase activity of MetCaCb (lines 156-159). However, MetCaCb itself does not catalyze the hydrolysis of arginine (line 167). Thus, reasoning regarding the mechanism of metformin hydrolysis based on the behavior of the enzyme when adding ABH needs more support. Could the authors provide further evidence, either computational or experimental, to further support these claims?

We agree that it is too preliminary to hypothesize the mechanism of the enzyme based on the current evidences. Therefore, the sentence “indicating the possible formation of a tetrahedral intermediate during MetCaCb-catalyzed metformin hydrolysis” has been removed. Nonetheless, we are still pretty confident in the observation that ABH inhibits the activity of MetCaCb, and hope this information is available for the interest of

readers.

(8) Asparagine is an unusual histidine-substituting proton-switching residue as presented here. The authors suggest that N199 in MetCa functions in this way (lines 237-240). It would be desirable to provide several examples from the literature where asparagine acts similarly?

A comparable case is the H141 of human arginase I (hARGI). The H141N mutant of hARGI retains significant activity, but the H141A mutant does not. This difference can be explained by their similar property of His and Asn: both can act as proton donor to E277 and as a proton acceptor from substrate arginine, resulting in the stabilization of the reaction intermediates [10]. The reference has been cited in the text accordingly (line 246). The sentence has been modified to: “The activity of the N199H variant was significantly decreased, and the activity of the N199A variant is completely lost (Supplementary Table 1). The residue N199 likely plays a role similar to its equivalent histidine in arginases which is used for stabilizing the reaction intermediate.” (Lines 242-246)

(9) Would it be possible to study whether the N199H mutant maintains its catalytic activity?

The activity of N199H is measured during revision and we included this in Supplementary Table 1. The result has been stated in the MS: “The activity of the N199H variant was significantly decreased, and the activity of the N199A variant is completely lost (Supplementary Table 1).” (Lines 242-244)

(10) Lines 109-113 mention that no active enzyme complex is obtained by mixing and incubating individually expressed and purified MetCa and MetCb together. However, an active enzyme can be obtained when the two are co-expressed. This would imply that either the heteromultimer cannot form after mixing, or that it does form but assumes an incorrect fold. Have you tried to measure the affinity of the individually/separately expressed subunits to each other? Were you able to determine that the individually expressed MetCa and MetCb proteins have the same monomeric fold as in the complex? We observed that the purified MetCb formed a stable homohexamer structure, as evidenced by SEC analysis. This could prevent the formation of heteromeric MetCaCb complex by mixing and incubating individually expressed and purified MetCa and MetCb together. Actually, in order to test the interaction of MetCa and MetCb in vitro, we had tried to perform a pull-down analysis using individually expressed His tagged-MetCa and tag free MetCb, however, no binding was observed. Instead, the interaction of MetCa and MetCb was observed only when MetCaCb was co-expressed and co-purified, as shown in Fig. 2a. The above clues indicate that individually purified MetCa and MetCb can't form an active heteromeric state, as the interaction surfaces required for hetero-multimeric association might be occupied by a preformed homomultimeric form of MetCb. We have now stated this in the discussion: “Metformin hydrolase activity cannot be detected when mixing individually purified MetCa and MetCb in vitro (Fig. 1c), indicating the failure of forming MetCaCb complex. This could be due

to the purified MetCb formed a well-organized homohexamer structure in solution, the interaction surfaces required for hetero-multimeric association might be occupied by a preformed homomultimeric form of MetCb.” Lines 336-341

(11) Please revise the sentence starting with “Comparable activity was detected...” (line 99), as it is hard to parse.

The sentence has been revised into “Coexpression of the two metal chaperones MetAB with MetCaCb had no significant influence on the activity (Fig. 1b)” Lines 99-100.

(12) Fig. 3 panel e would be improved if the labeled residues were highlighted with a different color and/or if the sticks were made thicker. In Extended Data Fig. 6, it is difficult to see both the stick representation of the structural elements and the residue labels, please modify.

The figures (Fig. 3e and Supplementary Fig. 6) have been modified as suggested.

1. Di Costanzo, L., et al., *Crystal structure of human arginase I at 1.29-Å resolution and exploration of inhibition in the immune response*. Proc Natl Acad Sci U S A, 2005. **102**(37): p. 13058-63.
2. Famakinwa, O., et al., *Biochemical characterization of a purified arginase from the gut of Oryctes rhinoceros larvae*. Ife Journal of Science, 2023. **25**: p. 137-146.
3. Funck, D., et al., *Discovery of a Ni²⁺-dependent guanidine hydrolase in bacteria*. Nature, 2022.
4. Pereira-Leal, J.B., et al., *Evolution of protein complexes by duplication of homomeric interactions*. Genome Biology, 2007. **8**(4): p. R51.
5. Hirsch-Kolb, H., H. Kolb, and D. Greenberg, *Nuclear Magnetic Resonance Studies of Manganese Binding of Rat Liver Arginase*. The Journal of biological chemistry, 1971. **246**: p. 395-401.
6. Uribe, E., et al., *Functional analysis of the Mn(2+) requirement in the catalysis of ureohydrolases arginase and agmatinase - a historical perspective*. J Inorg Biochem, 2020. **202**: p. 110812.
7. Olson, J., N. Mehta, and R. Maier, *Requirement of nickel metabolism proteins HypA and HypB for full activity of both hydrogenase and urease in Helicobacter pylori (vol 39, pg 176, 2001)*. Molecular microbiology, 2001. **39**: p. 176-82.
8. D'Antonio, E.L., Y. Hai, and D.W. Christianson, *Structure and function of non-native metal clusters in human arginase I*. Biochemistry, 2012. **51**(42): p. 8399-409.
9. Hernández, V.M., A. Arteaga, and M.F. Dunn, *Diversity, properties and functions of bacterial arginases*. FEMS Microbiol Rev, 2021. **45**(6).
10. Velázquez-Libera, J.L., et al., *On the Nature of the Enzyme–Substrate Complex and the Reaction Mechanism in Human Arginase I. A Combined Molecular Dynamics and QM/MM Study*. ACS Catalysis, 2020. **10**(15): p. 8321-8333.

Reviewers' Comments:

Reviewer #1:

Remarks to the Author:

Dear Authors,

Thank you for these revisions. From the first time I reviewed this paper, I was excited by the scientific results and implications. This is beautiful work describing the microbial degradation of metformin. I will proceed addressing the work, point-by-point, but overall recommend this article to be accepted. My comments are seen in red below. Please note that all Reviewer 1's minor comments were addresses, and thus are not specifically addressed below.

Reviewer #1:

In this manuscript, Li et al characterize a fascinating enzymatic system that enables bacterial catabolism of the common medication metformin via a metal-dependent hydrolase forming guanylurea and dimethylamine. This is a really beautiful paper and enzymatic system. Not only does MetCaCb present distinctive chemical reactivity, but also the unique evolution of two separate genes encoding arginase proteins (one catalytic, and one non-catalytic), both required for enzymatic catalysis. The evolution and mechanism of this metformin hydrolase could have great impact on understanding microbial catabolism in the ancient arginase hydrolase family, and potentially contribute to our understanding of metformin is a prominent drug, particularly in diabetes type 2 treatment.

Overall I recommend this manuscript for publication with minor revisions; these data present exciting implications for both metformin catabolism and for the wider arginase protein family. These remarks stand.

Recommendations:

(1) One of the most intriguing parts of this paper, and point emphasized in this paper, is the impact that this catabolic pathway might have on gut microbial catabolism in human hosts who are administrated metformin. Authors use the Enzyme Function Initiative (EFI) Enzyme Similarity Tool (EST) to form a sequence similarity network (SSN) (see Extended Data 7). Using the EFI's Computationally-Guided Functional Profiling Tool (CGFP) to observe species that encode for these genes in the human microbiome would be highly valuable. EST-CGFP is a very user-friendly, publicly available resource that searches human metagenomic databases for prominence of these genes encoded in the human microbiome—running an analysis to see how abundant your genes of interest are in the human microbiome (particularly gut) would strengthen the impact of this paper. If these enzymes are abundant in human microbial species, this would be a compelling piece of data (possibly even to include in main figures).

Thank you for your pertinent suggestion, and we have now performed such analysis using the EFI-CGFP tool. It turned out that no metformin hydrolase homolog was identified in over 380 metagenomes from the human microbiome database. Moreover, as our analysis stated in the first version (Fig. 4A), we also didn't observe any hits originating from human microbiome when searching for the clustered metformin hydrolase genes in the NCBI database. We have now included this information in the revised MS (lines 372-373). We will keep performing this search from time to time.

Although the metformin hydrolase seems not to be currently present in the human microbiome, the emerging of metforminase activity in bacteria concerns us as well, at least from the following aspects: i) the potential evolution of metforminase from a distinct genetic origin (i.e., isoenzyme) in specific environment and selective pressure in the human gut; ii) the possible dispersion of metforminase genes, such as the metCaCb identified here, from the environmental microbes to human gut microbiome.

Thank you for looking into this aspect—I do not think the results withdraw any aspect of the paper. Regardless of the results this metabolic pathway is interesting and impactful; I was just curious about this question and its answer provides preliminary insight into future studies of metformin-human microbiome interactions.

(2) I noticed how optimal catalytic temperatures are $\sim 50^{\circ}\text{C}$ and pH ~ 9.0 . These are notably different than human body temp ($\sim 37^{\circ}\text{C}$ and closer to neutral, or even acidic pHs). What are thoughts of human biological relevance keeping in mind these experimental results?

It is prevalent that arginase family enzymes prefer reaction conditions that are alkaline and around $50\text{-}60^{\circ}\text{C}$, including human arginases [1] and arginases from gut microbes [2]. It seems that these enzymes function under physiological conditions, although they probably never operate near their optimum conditions. The metformin hydrolase has been identified from bacteria found in wastewater from urban domestic sewage treatment plants, and we haven't observed its homolog in the human gut microbes as aforementioned. Therefore, this enzyme most likely works in environments such as water, soil, etc, where the strain was isolated in this case.

Could possibly address this in the manuscript (I think many people, including myself, love to think about how microbes prominent in soil/marine/etc microbiota might eventually play a role in mammalian microbiome systems. This could be a place to discuss such ideas.

(3) Do you still have crystals, and if so, could you collect anomalous data at the Ni K-edge ($\sim 8330\text{ eV}/1.49\text{ \AA}$)? This would strength argument of this enzymes specifically being Ni vs other metal (such as Mn dependent). What might be a competitive advantage of using Ni vs other more abundant metals in the cell?

Thank you for your constructive suggestion. Although we no longer have crystals, we repeated the crystallization experiment during this revision and obtained new ones for collecting anomalous data at 8347 eV . Unfortunately, the diffraction data are challenging for further analysis due to either weak signal or some kind of twin crystal. In order to determine the Ni content of the MetCaCb, we also repeated the ICP-MS analysis using the MetCaCb purified from the *E. coli* cells grown in nickel-rich medium rather than LB medium previously used, resulting in an observation of 3.9 nickel atoms per $\alpha_2\beta_4$ hexamer of MetCaCb, consistent with the di-nickel center. Under physiological conditions, the intracellular nickel concentration is low and the load of nickel onto MetCaCb is assisted by the nickel incorporation-associated proteins (MetA and MetB), as evidenced by the inability of the metAB knockout strain to grow with metformin (Fig. 1e). Taken together, these results further support the assertion that the metformin hydrolase is a di-nickel enzyme. Most arginase family enzymes prefer manganese for their catalysis; however, the metformin and guanidine hydrolases [3] are the only two members in this family which are activated by nickel. Both substrates metformin and guanidine have a more resonance stabilized structure compared to the arginine, the substrate of arginases. Therefore, the nickel-dependency of metformin and guanidine hydrolases is likely due to the higher Lewis acidity of nickel than manganese, which is likely required for catalyzing the hydrolysis of metformin and guanidine. These discussions have been included in the revised MS (lines 297-301).

This explanation and additions to the text are helpful. In the future, X-ray fluorescence studies (to gain insight into metals in protein crystal regardless of protein crystal) might help with metal identification and further support study, but I do not think such studies are necessary to publish this article. I think this article is impactful without these additional studies.

(4) It is so cool how MetCa and MetCb are so similar, both by sequence and overall architecture. It could be helpful to describe in text the residues that specifically dictate the 2:1 stoichiometry. Are there conserved binding residue pairs that enable this binding and the importance of the non-catalytic MetCb subunit?

Thank you for the constructive suggestions that led us to ruminate deeper into the interaction and evolutionary scenario of MetCa and MetCb. As suggested, the interacting residues of MetCa and MetCb were analyzed. While the residues dictating the binding of heterologous multimer seem currently enigmatic, a large number of mutation analyses would be required to determine if certain residues influence the heterologous binding. So far, we can propose that the N-terminal tail of MetCb, located at the interface of three subunits ($\alpha_1\beta_1\beta_4$ or $\alpha_2\beta_2\beta_3$) (Supplementary Fig. 5b), is crucial for the heterologous binding. Notably, truncating the 23 amino acids in the N-terminal of MetCb resulted in a failure to achieve soluble expression of the MetCaCb complex, suggesting its contribution to the formation of the heteromultimeric protein complex. These analyses have been strengthened in the MS (lines 230-233).

Thank you for the thoughtful consideration—I believe this insight strengthens the paper.

Is there any speculation as to why MetCa and MetCb would have evolved as this heterooligomer vs homooligomer? Authors suggest that this evolved via a gene duplication event—how conserved is this among species, and what might be a competitive advantage that would allow this heterooligomeric species to evolve, distinct from other members of the arginase enzymes.

Based on the sequences, structures, and gene organization of MetCa and MetCb, we hypothesized that MetCaCb evolved from a common ancestor, likely encoding a homomeric protein, through a gene duplication event and subsequent divergences, resulting in heteromeric complexes. The evolution of the heterooligomer appears to precede the emergence of metformin hydrolase activity, as evidenced by the lack of activity or extremely low activity of the heterohexameric homologs of MetCaCb (Fig. 4). Efficient metformin hydrolases have only been identified in *Aminobacter* spp. and *Pseudomonas* spp., indicating that the metformin hydrolase offers distinct advantages specific to these microorganisms.

Generally, the duplication of a gene encoding for an oligomeric protein can lead to numerous possible outcomes, and the changes of oligomeric states of descendant homologs are always associated with functional innovations. Compared to the homomeric state, the evolution of heterooligomeric complexes might confer functional versatility by allowing for more intricate regulation and fine-tuning of catalytic efficiency or substrate specificity, potentially enhancing the organism's ability to adapt to diverse metabolic demands or environmental challenges. For homomeric enzymes with catalytic sites present in each of subunits, such as arginase family enzymes, the scope of evolving non-catalytic regulation is relatively limited because each subunit is constrained to retain the primary function. However, for heteromeric enzymes, since two or more proteins are mediating the primary function, selection pressure may relax and allow the emergence of secondary functions such as structural scaffold. This opens numerous possible routes of functional innovation. Although heteromers are less common in prokaryotic proteomes than eukaryotic proteomes [4], the MetCaCb would be a sophisticated example of heteromeric enzyme in bacteria, and the story of the MetCaCb evolution would be of great interest in further investigation.

I agree this is super interesting for future investigation—thank you for the consideration of this point. I agree this point has been addressed at this point and leaves room for future exciting research.

(5) You discuss how MetCb proteins do not have residues to enable di-metal coordination. Are these mutations conserved across species, and are all those genes co-localized with a gene like MetCa in the genome? Also it would be helpful to show crystallographic composite omit maps to convince reader that metal is not bound in this extremely similar protein—this is an extremely cool system but I want you all to strength the argument of how/why MetCb might have evolved and whether it's conserved among other members of the family, possibly associated with other arginase enzymes.

We analyzed the MetCb homologs containing mutations in di-metal coordinating residues (Fig. 4). These proteins exhibit AA sequence identities of 58%-100% (coverage >97%) to MetCb. All of these protein-encoding genes are co-localized with a MetCa-like gene and are distributed in the phyla of bacteria including the Proteobacteria, Actinomycetota and Acidobacteriota (Fig. 4). The MetCb-like proteins from *Aminobacter* and *Pseudomonas* contain five identical mutations in di-metal coordinating residues to MetCb, while three or four mutations present in others (Fig. 4). The crystallographic composite omit maps have been included in the modified Supplementary Fig. 5c, d, clearly indicating that no metal ion is bound in MetCb.

These additional figures and insight are helpful. Thank you.

(6) You all discuss the N- and C-terminal tails in MetCa and MetCb and how those might regulate oligomeric state. These regions look relatively flexible (at least in Fig 3b). What are the B-factors in these regions? Are the orientation similar in the crystal vs EM structures? Could those orientations in the crystal structures possibly be held in place by crystallographic contacts? The average B-factor of the discussed N- and C-terminal tails in MetCa and MetCb has been

indicated in the text (lines 198 and 230) and colored in the Supplementary Fig. 5b. The N-terminal tails of MetCa are stable which contributes to the interaction of the two MetCa subunits. The four MetCb subunits adopt two types of conformation, i.e., $\beta 1\beta 3$ in one conformation and $\beta 2\beta 4$ in another (Fig. 3a), and the N- and C-terminal tails of them seem to exhibit different flexibility. The electron density of N- and C-terminal tails can only be seen in the $\beta 1\beta 3$ subunits, whereas they are missing in the $\beta 2\beta 4$ subunits. This could be due to that the interaction of N- and C-terminal tails of $\beta 1\beta 3$ with the MetCa subunits stabilized their conformation. Although we are unable to distinguish the orientation in the CryoEM map due to the limitation of resolution, and cannot rule out the possibility that the crystallographic contacts contribute to the observed conformation of the N- and C-terminal tails in MetCa and MetCb, it should be noted that: 1) the N- and C-terminal tails of the $\beta 1\beta 3$ subunits exhibit significant polar contacts with MetCa (Supplementary Fig. 5b); 2) Truncation of these tails results in a failure to achieve soluble expression of the MetCaCb complex. Therefore, it is most likely that the interaction of the flexible tails of MetCb with MetCa exists in solution state and plays an important role in the heterohexameric architecture. These additional figures and insight are helpful. Thank you.

(7) Authors mention residues involved in substrate binding (for example S289, page 12, line 239). How do you know this residue is involved with binding? If only from computational docking, I would make it clear how these are hypotheses. Is it conserved among species? Similarly authors state how residues such as N199 (similar to an equivalent His) are involved with proton shuffling. Be clear on how these conclusions are being drawn, and if the basis is solely on computational docking. Also a figure with a Chemdraw proposed mechanism would help readers interpret such statements. If authors are unable to get a substrate (or analog) bound structure, comparison to other members of the enzyme family and their proposed substrate binding and mechanisms would be helpful.

We agree that the currently evidences are too preliminary to elucidate the catalytic and binding details of this enzyme, and the statements have been rewritten: "The equivalent residue of N199 in the arginase family enzymes is a conserved noncoordinating histidine which is catalytically important during the reaction process. The activity of the N199H variant was significantly decreased, and the activity of the N199A variant is completely lost (Supplementary Table 1). The residue N199 likely plays a role similar to its equivalent histidine in arginases which is used for stabilizing the reaction intermediate. Additionally, the substrate pocket residue S289 and E320 was also essential for metformin hydrolase activity (Supplementary Table 1)."

Thank you for this clarification in the text. This is very helpful.

(8) Along with thinking about mechanism, how is Ni redox state reset and why do you think this enzyme might be Ni-specific from a mechanistic standpoint? Can you gain insight for hypotheses based on other members of the enzyme family?

The nickel enzymes include redox enzymes and nonredox hydrolases. For the Ni redox enzymes, the Ni ion exhibits versatility in redox properties, exploiting the +3/+2 states (superoxide dismutase and hydrogenase), the +2/+1 states (hydrogenase, CO dehydrogenase, and acetyl-CoA synthase), and perhaps even reaching the 0 state (acetyl-CoA synthase) for the redox enzymes. On the other hand, the nonredox Ni hydrolases (e.g., urease) remain in the +2 state during catalysis. The metformin hydrolase apparently belongs to the latter, which appears to only use Ni²⁺ as a Lewis acid. Most arginase family enzymes prefer manganese for their catalysis; however, metformin and guanidine hydrolases [3] are the only two members in this family activated by nickel. Both metformin and guanidine have a more resonance-stabilized structure compared to arginine, the substrate of arginase. Therefore, the nickel-dependency of metformin and guanidine hydrolases would be due to that the Lewis acidity of nickel is higher than manganese, likely required for catalyzing the hydrolysis of metformin and guanidine. These discussions have been included in the revised MS (lines 296-300).

Again, these additions are very helpful and alleviate concern. Thank you.

[Please note that all minor corrections have been addressed]

Overall I loved reading this paper. It is very clear and a really fascinating enzyme system (especially as it requires two similar yet distinct subunits). Clarifications and additions discussed above would be helpful, but overall I am very excited for this work to be published and recommend it for publication upon minor revisions and additions.

Reviewer #3 (Remarks to the Author):

The manuscript by Li et al. deals with the bacterial degradation of metformin, an anthropogenic biguanidine derivative. It is used therapeutically, particularly for the treatment of type 2 diabetes, and is one of the most frequently released active pharmaceutical compounds worldwide. Bacterial degradation by aquatic and soil bacteria is considered an important process in the removal of metformin from the environment. In recent years, metformin-degrading bacteria have been isolated and gene clusters and enzymes involved in metformin degradation have been identified. Only the first enzyme of the degradation pathway, which cleaves metformin into guanylurea and dimethylamine, has not yet been studied in detail.

In the present manuscript by Li et al., metformin hydrolase was produced heterologously from an *Aminobacter* sp. for the first time and characterized kinetically, biochemically, structurally (X-ray crystallography and cryo-EM) and phylogenetically. The hydrolase from the arginase protein family is a di-nickel-center-containing heterohexameric enzyme ($\alpha_2\beta_4$), of which only the α -subunit carries a catalytic center. However, the enzyme is only active in the presence of both subunits. A phylogenetic analysis identified further potential metformin-cleaving enzymes, some of which were heterologously produced in active (albeit weak) form.

Overall, it is a very comprehensive, solid work in which the first enzyme of metformin degradation was proven beyond doubt. The significance of the work for the field of research stems from the widespread use and distribution of metformin. Although the hydrolase itself is unusual in its subunit composition, the catalyzed reaction is rather trivial and typical for members of the well-studied ureido-hydrolase family. Overall, the work is very well written, and also scientifically at a very high level.

Major points:

(1) In particular, the figures and their captions are too small and hard to read, at least in the form I have. The authors should check the legibility of all illustrations (contrast of color, font size, overall size...).

We apologize for this inconvenience. We have modified the figures and their captions throughout in order to achieve more legibility.

This issue has been addressed.

(2) In the discussion, one could elaborate on the aspect that metformin is a xenobiotic that was first introduced into the environment by humans. It is therefore not surprising that metformin hydrolase is not a particularly efficient enzyme (e.g. very high K_m value). Also those enzymes shown in Fig. 4c are, if at all, only extremely poor metformin hydrolases. So it seems that the time since the introduction of metformin has been somewhat very short to evolve an efficient enzyme. We agree and extend the discussion for including this point in the revised MS. It has now been stated in the discussion: "Metformin has been extensively synthesized and used by human to treat diabetes and subsequently has been found in the environment for only a few decades. Our results revealed that a set of arginase family enzymes found in the past few years from environmental bacteria are rapidly evolved to become active hydrolases against metformin. The metformin hydrolase is not a particularly efficient enzyme with a relative high K_m value and low catalytic efficiency (Supplementary Table 1). So it seems that the time since the introduction of metformin into the environment has been too short to evolve an efficient enzyme." Lines 362-369.

This discussion is highly valuable and has been discussed and addressed.

Minor points:

(1) Fig. 2b: indicate in the legend the symbol for activity. The authors should use a common style of giving activities. In Fig 1a, a rate, in Fig. 2b, a final product concentration is given. At least, the

authors should indicate after which incubation time the product was formed. The activities are indicated by reaction rate throughout except the Fig.2b. In Fig.2b, we aimed to determine the active fractions from SEC rather than to quantify the rate of the enzymatic reaction. Therefore, the activities of each fraction are indicated by the production of guanylylurea. We have now indicated the reaction conditions in the figure legend for Fig. 2b. I feel this point has been addressed.

(2) Suppl. Fig 1ab: the authors should describe what is shown in the lanes that were cropped. Descriptions for all lanes in the uncropped SDS-PAGE images have been added in the legends, now it is in supplementary Fig. 8 and 9. This point has been addressed.

(3) Ext. Fig. 2 and throughout the manuscript. Instead of using N.D., the authors should rather state that the concentration is under the detection limit, and give a number for the latter (i.e. <0.01 mM). We have given a number for the detection limit accordingly in the figure legends. This point has been addressed.

(4) L.752 replace 'with' a concentration by 'at' a concentration. Corrected. Supplementary Figure 2 legend. This point has been addressed.

(5) L.171: replace 'recognition' by 'specific binding'. Corrected. Line 169. This point has been addressed.

(6) Fig. 3a. It is difficult to spot the metals (green spheres) in the printed version: too small. Fig 3c, for labelling of amino acids, use colours with more contrast. The amino acids in Fig. 3e can hardly be seen. The figures have been improved as suggested. This point has been addressed.

(7) L.180: ...was roughly inferred.. better: ... is reflected by the different.... or ...explains the different.... Modified. Line 178. This point has been addressed.

(8) Ext. Fig 4a: It should be zoomed in to better recognize the micrograph. For better clarity, micrographs with different magnifications have been provided in the revised Supplementary Figure 4a. This point has been addressed.

(9) L.210 ff.: this is not clear to me: the enzyme as isolated had almost no Ni, but Mn, which explains the low activity (Fig. 1b)?? The authors may place metal content analysis better at the beginning, because it fits more to the reconstitution assays shown in Fig. 1b. In Fig. 1b, we tested the activities of metformin hydrolase with crude enzymes, and we observed very low activity when no extra metal ion or manganese was added to the reaction mixtures. This could be due to the influence of a trace of nickel in the LB media used to grow the cells. Because analysis of the metal ion content of recombinant MetCaCb purifying from the LB-grown cells indicated that MetCaCb contains 0.8 manganese and 0.05 nickel atoms per $\alpha 1\beta 2$ trimer. Addition of manganese into the reaction mixture can't significantly increase the activity of MetCaCb, as the enzyme needs nickel to activate the reaction, as outlined in the response to points 3 and 8 by reviewer 1. In our original analysis, the enzyme was purified from LB media-grown E. coli cells without the addition of any extra metal ion, resulting in the observation of 0.8 manganese and 0.05 nickel

atoms per $\alpha 1\beta 2$ trimer. We noted that this could be due to the limited content of nickel in the LB media. During revision, we performed an ICP-MS analysis for the recombinant MetCaCb purified from cells grown with nickel-rich LB media, and 1.95 nickel atoms per $\alpha 1\beta 2$ trimer were detected. We have now revised the MS accordingly. (lines 214-218)

This point has been addressed and caveats clarified.

(10) Comment on the low activity of heterologously produced enzymes, do you believe that they prefer other substrates

We agree that these low-activity homologous enzymes may well prefer other substrates; for example, the derivatives of guanidine or biguanide. Here, we focus on the characterization of a metformin hydrolase. However, in the future, it will be intriguing to investigate the substrate preference of metformin hydrolase and its homologs, as well as their evolutionary connections. I think this is a fair analysis and does not need broader discussion—this study addresses generalized metformin metabolism, which is valuable basic biological research. It is important to understand how mechanistic metformin metabolism studies, such as those in this paper, can have broad implications in how this widespread drug is catabolized both within the environment and in mammalian microbiomes.

Overall this is an exciting paper that I highly recommend for publication.

Reviewer #2:

Remarks to the Author:

The authors describe the characterization of a heteromeric metforminase that is encoded in the genomes of several recently identified metformin-utilizing bacteria. Metformin is one of the most widely prescribed drugs and is a major anthropogenic contaminant of waterways worldwide. Thus, characterization of metforminase is very interesting to a wide audience. The manuscript is well-written and the work is very high quality.

My major comment was that the authors could strengthen the claim that two Ni chaperones (metAB) are required in vivo by showing that expression of metforminase (metCaCb) occurs in the metAB mutant. The authors used a complementation assay to show that expression of metforminase is indeed unaffected in the metAB mutant. This result is very convincing.

The science and methodology is sound. I have no further comments.

REVIEWERS' COMMENTS TO FIRST REVISION

Reviewer #1 (Remarks to the Author):

Dear Authors,

Thank you for these revisions. From the first time I reviewed this paper, I was excited by the scientific results and implications. This is beautiful work describing the microbial degradation of metformin. I will proceed addressing the work, point-by-point, but overall recommend this article to be accepted. My comments are seen in red below.

Please note that all Reviewer 1's minor comments were addresses, and thus are not specifically addressed below.

Thank you for your constructive comments on our manuscript.

Reviewer #1:

In this manuscript, Li et al characterize a fascinating enzymatic system that enables bacterial catabolism of the common medication metformin via a metal-dependent hydrolase forming guanylyurea and dimethylamine. This is a really beautiful paper and enzymatic system. Not only does MetCaCb present distinctive chemical reactivity, but also the unique evolution of two separate genes encoding arginase proteins (one catalytic, and one non-catalytic), both required for enzymatic catalysis. The evolution and mechanism of this metformin hydrolase could have great impact on understanding microbial catabolism in the ancient arginase hydrolase family, and potentially contribute to our understanding of metformin is a prominent drug, particularly in diabetes type 2 treatment.

Overall I recommend this manuscript for publication with minor revisions; these data present exciting implications for both metformin catabolism and for the wider arginase protein family.

These remarks stand.

Recommendations:

(1) One of the most intriguing parts of this paper, and point emphasized in this paper, is the impact that this catabolic pathway might have on gut microbial catabolism in human hosts who are administrated metformin. Authors use the Enzyme Function Initiative (EFI) Enzyme Similarity Tool (EST) to form a sequence similarity network (SSN) (see Extended Data 7). Using the EFI's Computationally-Guided Functional Profiling Tool (CGFP) to observe species that encode for these genes in the human microbiome would be highly valuable. EST-CGFP is a very user-friendly, publicly available resource that searches human metagenomic databases for prominence of these genes encoded in the human microbiome—running an analysis to see how abundant your genes of interest are in the human microbiome (particularly gut) would strengthen the impact of this paper. If these enzymes are abundant in human microbial species, this would be a compelling piece of data (possibly even to include in main figures).

Thank you for your pertinent suggestion, and we have now performed such analysis using the EFI-CGFP tool. It turned out that no metformin hydrolase homolog was

identified in over 380 metagenomes from the human microbiome database. Moreover, as our analysis stated in the first version (Fig. 4A), we also didn't observe any hits originating from human microbiome when searching for the clustered metformin hydrolase genes in the NCBI database. We have now included this information in the revised MS (lines 372-373). We will keep performing this search from time to time.

Although the metformin hydrolase seems not to be currently present in the human microbiome, the emerging of metforminase activity in bacteria concerns us as well, at least from the following aspects: i) the potential evolution of metforminase from a distinct genetic origin (i.e., isoenzyme) in specific environment and selective pressure in the human gut; ii) the possible dispersion of metforminase genes, such as the metCaCb identified here, from the environmental microbes to human gut microbiome.

Thank you for looking into this aspect—I do not think the results withdraw any aspect of the paper. Regardless of the results this metabolic pathway is interesting and impactful; I was just curious about this question and its answer provides preliminary insight into future studies of metformin-human microbiome interactions.

(2) I noticed how optimal catalytic temperatures are $\sim 50^{\circ}\text{C}$ and pH ~ 9.0 . These are notably different than human body temp ($\sim 37^{\circ}\text{C}$ and closer to neutral, or even acidic pHs). What are thoughts of human biological relevance keeping in mind these experimental results?

It is prevalent that arginase family enzymes prefer reaction conditions that are alkaline and around $50\text{-}60^{\circ}\text{C}$, including human arginases [1] and arginases from gut microbes [2]. It seems that these enzymes function under physiological conditions, although they probably never operate near their optimum conditions. The metformin hydrolase has been identified from bacteria found in wastewater from urban domestic sewage treatment plants, and we haven't observed its homolog in the human gut microbes as aforementioned. Therefore, this enzyme most likely works in environments such as water, soil, etc, where the strain was isolated in this case.

Could possibly address this in the manuscript (I think many people, including myself, love to think about how microbes prominent in soil/marine/etc microbiota might eventually play a role in mammalian microbiome systems. This could be a place to discuss such ideas.

These contents have been included in the manuscript as suggested. (Lines 379-378)

(3) Do you still have crystals, and if so, could you collect anomalous data at the Ni K-edge ($\sim 8330\text{ eV}/1.49\text{ \AA}$)? This would strength argument of this enzymes specifically being Ni vs other metal (such as Mn dependent). What might be a competitive advantage of using Ni vs other more abundant metals in the cell?

Thank you for your constructive suggestion. Although we no longer have crystals, we repeated the crystallization experiment during this revision and obtained new ones for collecting anomalous data at 8347 eV . Unfortunately, the diffraction data are challenging for further analysis due to either weak signal or some kind of twin crystal. In order to determine the Ni content of the MetCaCb, we also repeated the ICP-MS analysis using the MetCaCb purified from the E. coli cells grown in nickel-rich medium rather than

LB medium previously used, resulting in an observation of 3.9 nickel atoms per $\alpha_2\beta_4$ hexamer of MetCaCb, consistent with the di-nickel center. Under physiological conditions, the intracellular nickel concentration is low and the load of nickel onto MetCaCb is assisted by the nickel incorporation-associated proteins (MetA and MetB), as evidenced by the inability of the metAB knockout strain to grow with metformin (Fig. 1e). Taken together, these results further support the assertion that the metformin hydrolase is a di-nickel enzyme.

Most arginase family enzymes prefer manganese for their catalysis; however, the metformin and guanidine hydrolases [3] are the only two members in this family which are activated by nickel. Both substrates metformin and guanidine have a more resonance stabilized structure compared to the arginine, the substrate of arginases. Therefore, the nickel-dependency of metformin and guanidine hydrolases is likely due to the higher Lewis acidity of nickel than manganese, which is likely required for catalyzing the hydrolysis of metformin and guanidine. These discussions have been included in the revised MS (lines 297-301).

This explanation and additions to the text are helpful. In the future, X-ray fluorescence studies (to gain insight into metals in protein crystal regardless of protein crystal) might help with metal identification and further support study, but I do not think such studies are necessary to publish this article. I think this article is impactful without these additional studies.

(4) It is so cool how MetCa and MetCb are so similar, both by sequence and overall architecture. It could be helpful to describe in text the residues that specifically dictate the 2:1 stoichiometry. Are there conserved binding residue pairs that enable this binding and the importance of the non-catalytic MetCb subunit?

Thank you for the constructive suggestions that led us to ruminate deeper into the interaction and evolutionary scenario of MetCa and MetCb. As suggested, the interacting residues of MetCa and MetCb were analyzed. While the residues dictating the binding of heterologous multimer seem currently enigmatic, a large number of mutation analyses would be required to determine if certain residues influence the heterologous binding. So far, we can propose that the N-terminal tail of MetCb, located at the interface of three subunits ($\alpha_1\beta_1\beta_4$ or $\alpha_2\beta_2\beta_3$) (Supplementary Fig. 5b), is crucial for the heterologous binding. Notably, truncating the 23 amino acids in the N-terminal of MetCb resulted in a failure to achieve soluble expression of the MetCaCb complex, suggesting its contribution to the formation of the heteromultimeric protein complex. These analyses have been strengthened in the MS (lines 230-233).

Thank you for the thoughtful consideration—I believe this insight strengthens the paper.

Is there any speculation as to why MetCa and MetCb would have evolved as this heterooligomer vs homooligomer? Authors suggest that this evolved via a gene duplication event—how conserved is this among species, and what might be a competitive advantage that would allow this heterooligomeric species to evolve, distinct from other members of the arginase enzymes.

Based on the sequences, structures, and gene organization of MetCa and MetCb, we

hypothesized that MetCaCb evolved from a common ancestor, likely encoding a homomeric protein, through a gene duplication event and subsequent divergences, resulting in heteromeric complexes. The evolution of the heterooligomer appears to precede the emergence of metformin hydrolase activity, as evidenced by the lack of activity or extremely low activity of the heterohexameric homologs of MetCaCb (Fig. 4). Efficient metformin hydrolases have only been identified in *Aminobacter* spp. and *Pseudomonas* spp., indicating that the metformin hydrolase offers distinct advantages specific to these microorganisms.

Generally, the duplication of a gene encoding for an oligomeric protein can lead to numerous possible outcomes, and the changes of oligomeric states of descendant homologs are always associated with functional innovations. Compared to the homomeric state, the evolution of heterooligomeric complexes might confer functional versatility by allowing for more intricate regulation and fine-tuning of catalytic efficiency or substrate specificity, potentially enhancing the organism's ability to adapt to diverse metabolic demands or environmental challenges. For homomeric enzymes with catalytic sites present in each of subunits, such as arginase family enzymes, the scope of evolving non-catalytic regulation is relatively limited because each subunit is constrained to retain the primary function. However, for heteromeric enzymes, since two or more proteins are mediating the primary function, selection pressure may relax and allow the emergence of secondary functions such as structural scaffold. This opens numerous possible routes of functional innovation. Although heteromers are less common in prokaryotic proteomes than eukaryotic proteomes [4], the MetCaCb would be a sophisticated example of heteromeric enzyme in bacteria, and the story of the MetCaCb evolution would be of great interest in further investigation.

I agree this is super interesting for future investigation—thank you for the consideration of this point. I agree this point has been addressed at this point and leaves room for future exciting research.

(5) You discuss how MetCb proteins do not have residues to enable di-metal coordination. Are these mutations conserved across species, and are all those genes co-localized with a gene like MetCa in the genome? Also it would be helpful to show crystallographic composite omit maps to convince reader that metal is not bound in this extremely similar protein—this is an extremely cool system but I want you all to strength the argument of how/why MetCb might have evolved and whether it's conserved among other members of the family, possibly associated with other arginase enzymes.

We analyzed the MetCb homologs containing mutations in di-metal coordinating residues (Fig. 4). These proteins exhibit AA sequence identities of 58%-100% (coverage >97%) to MetCb. All of these protein-encoding genes are co-localized with a MetCa-like gene and are distributed in the phyla of bacteria including the Proteobacteria, Actinomycetota and Acidobacteriota (Fig. 4). The MetCb-like proteins from *Aminobacter* and *Pseudomonas* contain five identical mutations in di-metal coordinating residues to MetCb, while three or four mutations present in others (Fig. 4). The crystallographic composite omit maps have been included in the modified

Supplementary Fig. 5c, d, clearly indicating that no metal ion is bound in MetCb.

These additional figures and insight are helpful. Thank you.

(6) You all discuss the N- and C-terminal tails in MetCa and MetCb and how those might regulate oligomeric state. These regions look relatively flexible (at least in Fig 3b). What are the B-factors in these regions? Are the orientation similar in the crystal vs EM structures? Could those orientations in the crystal structures possibly be held in place by crystallographic contacts?

The average B-factor of the discussed N- and C-terminal tails in MetCa and MetCb has been indicated in the text (lines 198 and 230) and colored in the Supplementary Fig. 5b. The N-terminal tails of MetCa are stable which contributes to the interaction of the two MetCa subunits. The four MetCb subunits adopt two types of conformation, i.e., $\beta 1\beta 3$ in one conformation and $\beta 2\beta 4$ in another (Fig. 3a), and the N- and C-terminal tails of them seem to exhibit different flexibility. The electron density of N- and C-terminal tails can only be seen in the $\beta 1\beta 3$ subunits, whereas they are missing in the $\beta 2\beta 4$ subunits. This could be due to that the interaction of N- and C-terminal tails of $\beta 1\beta 3$ with the MetCa subunits stabilized their conformation. Although we are unable to distinguish the orientation in the CryoEM map due to the limitation of resolution, and cannot rule out the possibility that the crystallographic contacts contribute to the observed conformation of the N- and C-terminal tails in MetCa and MetCb, it should be noted that: 1) the N- and C-terminal tails of the $\beta 1\beta 3$ subunits exhibit significant polar contacts with MetCa (Supplementary Fig. 5b); 2) Truncation of these tails results in a failure to achieve soluble expression of the MetCaCb complex. Therefore, it is most likely that the interaction of the flexible tails of MetCb with MetCa exists in solution state and plays an important role in the heterohexameric architecture.

These additional figures and insight are helpful. Thank you.

(7) Authors mention residues involved in substrate binding (for example S289, page 12, line 239). How do you know this residue is involved with binding? If only from computational docking, I would make it clear how these are hypotheses. Is it conserved among species? Similarly authors state how residues such as N199 (similar to an equivalent His) are involved with proton shuffling. Be clear on how these conclusions are being drawn, and if the basis is solely on computational docking. Also a figure with a Chemdraw proposed mechanism would help readers interpret such statements. If authors are unable to get a substrate (or analog) bound structure, comparison to other members of the enzyme family and their proposed substrate binding and mechanisms would be helpful.

We agree that the currently evidences are too preliminary to elucidate the catalytic and binding details of this enzyme, and the statements have been rewritten: “The equivalent residue of N199 in the arginase family enzymes is a conserved noncoordinating histidine which is catalytically important during the reaction process. The activity of the N199H variant was significantly decreased, and the activity of the N199A variant is completely lost (Supplementary Table 1). The residue N199 likely plays a role similar to its equivalent histidine in arginases which is used for stabilizing the reaction

intermediate. Additionally, the substrate pocket residue S289 and E320 was also essential for metformin hydrolase activity (Supplementary Table 1).”

Thank you for this clarification in the text. This is very helpful.

(8) Along with thinking about mechanism, how is Ni redox state reset and why do you think this enzyme might be Ni-specific from a mechanistic standpoint? Can you gain insight for hypotheses based on other members of the enzyme family?

The nickel enzymes include redox enzymes and nonredox hydrolases. For the Ni redox enzymes, the Ni ion exhibits versatility in redox properties, exploiting the +3/+2 states (superoxide dismutase and hydrogenase), the +2/+1 states (hydrogenase, CO dehydrogenase, and acetyl-CoA synthase), and perhaps even reaching the 0 state (acetyl-CoA synthase) for the redox enzymes. On the other hand, the nonredox Ni hydrolases (e.g., urease) remain in the +2 state during catalysis. The metformin hydrolase apparently belongs to the latter, which appears to only use Ni²⁺ as a Lewis acid. Most arginase family enzymes prefer manganese for their catalysis; however, metformin and guanidine hydrolases [3] are the only two members in this family activated by nickel. Both metformin and guanidine have a more resonance-stabilized structure compared to arginine, the substrate of arginase. Therefore, the nickel-dependency of metformin and guanidine hydrolases would be due to that the Lewis acidity of nickel is higher than manganese, likely required for catalyzing the hydrolysis of metformin and guanidine. These discussions have been included in the revised MS (lines 296-300).

Again, these additions are very helpful and alleviate concern. Thank you.

[Please note that all minor corrections have been addressed]

Overall I loved reading this paper. It is very clear and a really fascinating enzyme system (especially as it requires two similar yet distinct subunits). Clarifications and additions discussed above would be helpful, but overall I am very excited for this work to be published and recommend it for publication upon minor revisions and additions.

Reviewer #3 (Remarks to the Author):

The manuscript by Li et al. deals with the bacterial degradation of metformin, an anthropogenic biguanidine derivative. It is used therapeutically, particularly for the treatment of type 2 diabetes, and is one of the most frequently released active pharmaceutical compounds worldwide. Bacterial degradation by aquatic and soil bacteria is considered an important process in the removal of metformin from the environment. In recent years, metformin-degrading bacteria have been isolated and gene clusters and enzymes involved in metformin degradation have been identified. Only the first enzyme of the degradation pathway, which cleaves metformin into guanylurea and dimethylamine, has not yet been studied in detail.

In the present manuscript by Li et al., metformin hydrolase was produced heterologously from an *Aminobacter* sp. for the first time and characterized kinetically,

biochemically, structurally (X-ray crystallography and cryo-EM) and phylogenetically. The hydrolase from the arginase protein family is a di-nickel-center-containing heterohexameric enzyme ($\alpha_2\beta_4$), of which only the α -subunit carries a catalytic center. However, the enzyme is only active in the presence of both subunits. A phylogenetic analysis identified further potential metformin-cleaving enzymes, some of which were heterologously produced in active (albeit weak) form.

Overall, it is a very comprehensive, solid work in which the first enzyme of metformin degradation was proven beyond doubt. The significance of the work for the field of research stems from the widespread use and distribution of metformin. Although the hydrolase itself is unusual in its subunit composition, the catalyzed reaction is rather trivial and typical for members of the well-studied ureido-hydrolase family. Overall, the work is very well written, and also scientifically at a very high level.

Major points:

(1) In particular, the figures and their captions are too small and hard to read, at least in the form I have. The authors should check the legibility of all illustrations (contrast of color, font size, overall size...).

We apologize for this inconvenience. We have modified the figures and their captions throughout in order to achieve more legibility.

This issue has been addressed.

(2) In the discussion, one could elaborate on the aspect that metformin is a xenobiotic that was first introduced into the environment by humans. It is therefore not surprising that metformin hydrolase is not a particularly efficient enzyme (e.g. very high K_m value). Also those enzymes shown in Fig. 4c are, if at all, only extremely poor metformin hydrolases. So it seems that the time since the introduction of metformin has been somewhat very short to evolve an efficient enzyme.

We agree and extend the discussion for including this point in the revised MS. It has now been stated in the discussion: "Metformin has been extensively synthesized and used by human to treat diabetes and subsequently has been found in the environment for only a few decades. Our results revealed that a set of arginase family enzymes found in the past few years from environmental bacteria are rapidly evolved to become active hydrolases against metformin. The metformin hydrolase is not a particularly efficient enzyme with a relative high K_m value and low catalytic efficiency (Supplementary Table 1). So it seems that the time since the introduction of metformin into the environment has been too short to evolve an efficient enzyme." Lines 362-369.

This discussion is highly valuable and has been discussed and addressed.

Minor points:

(1) Fig. 2b: indicate in the legend the symbol for activity. The authors should use a common style of giving activities. In Fig 1a, a rate, in Fig. 2b, a final product concentration is given. At least, the authors should indicate after which incubation time the product was formed.

The activities are indicated by reaction rate throughout except the Fig.2b. In Fig.2b, we

aimed to determine the active fractions from SEC rather than to quantify the rate of the enzymatic reaction. Therefore, the activities of each fraction are indicated by the production of guanylyurea. We have now indicated the reaction conditions in the figure legend for Fig. 2b.

I feel this point has been addressed.

(2) Suppl. Fig 1ab: the authors should describe what is shown in the lanes that were cropped

Descriptions for all lanes in the uncropped SDS-PAGE images have been added in the legends, now it is in supplementary Fig. 8 and 9.

This point has been addressed.

(3) Ext. Fig. 2 and throughout the manuscript. Instead of using N.D., the authors should rather state that the concentration is under the detection limit, and give a number for the latter (i.e. <0.01 mM)

We have given a number for the detection limit accordingly in the figure legends.

This point has been addressed.

(4) L.752 replace 'with' a concentration by 'at' a concentration

Corrected. Supplementary Figure 2 legend.

This point has been addressed.

(5) L.171: replace 'recognition' by 'specific binding'

Corrected. Line 169.

This point has been addressed.

(6) Fig. 3a. It is difficult to spot the metals (green spheres) in the printed version: too small. Fig 3c, for labelling of amino acids, use colours with more contrast. The amino acids in Fig. 3e can hardly be seen.

The figures have been improved as suggested.

This point has been addressed.

(7) L.180: ...was roughly inferred.. better: ... is reflected by the different.... or ...explains the different....

Modified. Line 178.

This point has been addressed.

(8) Ext. Fig 4a: It should be zoomed in to better recognize the micrograph

For better clarity, micrographs with different magnifications have been provided in the revised Supplementary Figure 4a.

This point has been addressed.

(9) L.210 ff.: this is not clear to me: the enzyme as isolated had almost no Ni, but Mn, which explains the low activity (Fig. 1b)?? The authors may place metal content

analysis better at the beginning, because it fits more to the reconstitution assays shown in Fig. 1b

In Fig. 1b, we tested the activities of metformin hydrolase with crude enzymes, and we observed very low activity when no extra metal ion or manganese was added to the reaction mixtures. This could be due to the influence of a trace of nickel in the LB media used to grow the cells. Because analysis of the metal ion content of recombinant MetCaCb purifying from the LB-grown cells indicated that MetCaCb contains 0.8 manganese and 0.05 nickel atoms per $\alpha 1\beta 2$ trimer. Addition of manganese into the reaction mixture can't significantly increase the activity of MetCaCb, as the enzyme needs nickel to activate the reaction, as outlined in the response to points 3 and 8 by reviewer 1.

In our original analysis, the enzyme was purified from LB media-grown E. coli cells without the addition of any extra metal ion, resulting in the observation of 0.8 manganese and 0.05 nickel atoms per $\alpha 1\beta 2$ trimer. We noted that this could be due to the limited content of nickel in the LB media. During revision, we performed an ICP-MS analysis for the recombinant MetCaCb purified from cells grown with nickel-rich LB media, and 1.95 nickel atoms per $\alpha 1\beta 2$ trimer were detected. We have now revised the MS accordingly. (lines 214-218)

This point has been addressed and caveats clarified.

(10) Comment on the low activity of heterologously produced enzymes, do you believe that they prefer other substrates

We agree that these low-activity homologous enzymes may well prefer other substrates; for example, the derivatives of guanidine or biguanide. Here, we focus on the characterization of a metformin hydrolase. However, in the future, it will be intriguing to investigate the substrate preference of metformin hydrolase and its homologs, as well as their evolutionary connections.

I think this is a fair analysis and does not need broader discussion—this study addresses generalized metformin metabolism, which is valuable basic biological research. It is important to understand how mechanistic metformin metabolism studies, such as those in this paper, can have broad implications in how this widespread drug is catabolized both within the environment and in mammalian microbiomes.

Overall this is an exciting paper that I highly recommend for publication.

Reviewer #2 (Remarks to the Author):

The authors describe the characterization of a heteromeric metforminase that is encoded in the genomes of several recently identified metformin-utilizing bacteria. Metformin is one of the most widely prescribed drugs and is a major anthropogenic contaminant of waterways worldwide. Thus, characterization of metforminase is very interesting to a wide audience. The manuscript is well-written and the work is very high quality.

My major comment was that the authors could strengthen the claim that two Ni chaperones (metAB) are required in vivo by showing that expression of metforminase (metCaCb) occurs in the metAB mutant. The authors used a complementation assay to show that expression of metforminase is indeed unaffected in the metAB mutant. This result is very convincing.

The science and methodology is sound. I have no further comments.
Thank you for your constructive comments on our manuscript.